# Learning-Augmented Algorithms for $k$-median via Online Learning

**Anish Hebbar**
Duke University
anishshripad.hebbar@duke.edu

**Rong Ge**
Duke University
rongge@cs.duke.edu

**Amit Kumar**
IIT Delhi
amitk@cse.iitd.ac.in

**Debmalya Panigrahi**
Duke University
debmalya@cs.duke.edu

## Abstract

The field of learning-augmented algorithms seeks to use ML techniques on past instances of a problem to inform an algorithm designed for a future instance. In this paper, we introduce a novel model for learning-augmented algorithms inspired by online learning. In this model, we are given a sequence of instances of a problem and the goal of the learning-augmented algorithm is to use prior instances to propose a solution to a future instance of the problem. The performance of the algorithm is measured by its average performance across all the instances, where the performance on a single instance is the ratio between the cost of the algorithm's solution and that of an optimal solution for that instance. We apply this framework to the classic $k$-median clustering problem, and give an efficient learning algorithm that can approximately match the average performance of the best fixed $k$-median solution in hindsight across all the instances. We also experimentally evaluate our algorithm and show that its empirical performance is close to optimal, and also that it automatically adapts the solution to a dynamically changing sequence.

## 1 Introduction

The field of learning-augmented algorithms has seen rapid growth in recent years, with the aim of harnessing the ever-improving capabilities of machine learning (ML) to solve problems in combinatorial optimization. Qualitatively, the goal in this area can be stated in simple terms: *apply ML techniques on prior problem instances to inform algorithmic choices for a future instance*. This broad objective has been interpreted in a variety of ways – online algorithms with *better competitive ratios* using ML predictions, *faster (offline) algorithms* based on ML advice, *better approximation factors* for (offline) NP-hard problems using ML advice, etc. This breadth of interpretation has led to a wealth of interesting research at the intersection of algorithm design and machine learning.

In this paper, we take inspiration from the field of online learning to propose a new model for learning-augmented algorithms. The stated goal of learning-augmented algorithms of using prior instances to inform the solution of a future instance bears striking similarity with online learning. In its simplest form, online learning comprises a sequence of instances in which the learner has to select from a set of actions to minimize unknown loss functions. The choice of the learner is based on the performance of these actions on prior instances defined by their respective loss functions. The goal is to optimize the performance of the learner in the long run – it aims to match the average performance of the best fixed action in hindsight. Viewing learning-augmented algorithms through the lens of online learning, we ask:

> *Can a learning-augmented algorithm compete in average performance over a sequence of problem instances with the best fixed solution in hindsight?*

39th Conference on Neural Information Processing Systems (NeurIPS 2025).

To instantiate this question, we consider the classic $k$-median problem in clustering. In this problem, we are given a set of $n$ points in a metric space, and the goal is to find $k$ *centers* such that the sum of distances of the points to their closest centers is minimized. We note that $k$-median is among the most well-studied problems in unsupervised learning. In the learning-augmented framework, consider an online sequence of $k$-median instances $V_1, V_2, \ldots, V_t, \ldots, V_T$ on some metric space $\mathcal{M}$. Now, suppose at time $t$, the learning-augmented algorithm $\mathcal{A}$ has access to the prior instances $V_1, V_2, \ldots, V_{t-1}$ and has to devise a solution comprising $k$ centers for the next instance $V_t$ (that is unknown to it). Once the algorithm $\mathcal{A}$ reveals its solution $Y_t$, the instance $V_t$ is revealed and the loss of the algorithm is the quality of the solution $Y_t$ for $V_t$, namely the ratio of the cost of $Y_t$ and that of an optimal solution for $V_t$.

We define the loss to be the ratio instead of the absolute cost because the instances that arrive can have very different scales. In particular, if one instance is very large and the rest are small, it suffices to perform well on just the large instance when looking at absolute cost. Thus, taking cost ratios ensures that the algorithm is incentivized to do well on every instance, regardless of size. Secondly, it is impossible to obtain regret bounds independent of the number of points $n$ when considering absolute costs. To see this, suppose we get a very large instance with essentially all $n$ points that is far away from all the previous (much smaller) instances which are cumulatively on $o(n)$ points. Any online learning algorithm would then incur a cost of $n\Delta$ for this single instance, where $\Delta$ is the aspect ratio of the metric space.[1] On the other hand, the benchmark solution can simply optimize for this large instance and incur small cost for this instance. It suffers a large cost on the previous instances, but these instances being cumulatively small (only $o(n)$ points), the overall cost of the benchmark remains small in terms of absolute cost.

Similar to online learning, $\mathcal{A}$ cannot hope to match optimal performance for a single instance, but across time, can it match the performance of the best fixed solution in hindsight? In other words, we would like to learn a competitive solution from prior instances when such a solution exists. In this paper, we show that this is indeed possible – for $k$-median clustering, we design an online learning algorithm that approximately matches the performance of the best fixed solution across any sequence of instances on a metric space.

A natural approach to the above goal would be to employ the tools of online learning directly by encoding each solution as an "expert" and running an experts' learning algorithm (see e.g., [Hazan, 2016]) on the problem instances. This does not work for two reasons, and these also reveal the main conceptual challenges in this problem. The first challenge is *computational*. The space of solutions to a discrete optimization problem such as $k$-median clustering is *exponentially large* and *non-convex*. Therefore, we cannot run experts' algorithms that maintain explicit weights over the entire set of solutions, neither can we use a black-box online convex optimization (OCO) algorithm (see e.g., [Hazan, 2016]). The second challenge is *information-theoretic*. Since the sequence of instances is revealed online, the learning algorithm does not know future instances when it proposes a solution. Therefore, the metric space itself, and the corresponding space of solutions, is *dynamically growing* over time. Indeed, the best solution in hindsight may not even be an available option to the algorithm at some intermediate time $t$. The main technical contribution of this paper is in overcoming this set of fundamental challenges to obtain a learning-augmented algorithm for $k$-median that matches the average performance of the best solution in hindsight.

Our algorithms have potential applications in clustering evolving data streams, which has been identified as an important open challenge in data stream mining research [Krempl et al., 2014]. Some examples include aggregating similar news stories for recommendations downstream [Gong et al., 2017] and maintaining accurate clustering of sensor data that fluctuates due to environmental conditions [Wang et al., 2024]. Another application lies in clustering network traffic where patterns of network user connections change slowly over time in normal circumstances, but can also undergo sudden shifts if a malicious attack occurs [Gong et al., 2017, Wang et al., 2024, Feng, 2006].

## 1.1 Our Contributions

**Problem Model and Definition.** Our first contribution is the new learning-augmented model that we propose in this paper. We define it in the context of the $k$-median problem, and call it LEARN-

---

[1]The aspect ratio of a metric space is the ratio of the maximum to minimum pairwise distance between distinct points in the metric space.

MEDIAN. The problem setting comprises an online sequence of instances of $k$-median defined on a metric space $\mathcal{M} = (V, d)$ with a set of $n$ points $V$ and distance function $d : V \times V \to \mathbb{R}_{\geq 0}$. The algorithm does not assume knowledge of the metric space $(V, d)$, its size $n$, or the length of the time horizon $T$ upfront. The instance at time $t$ comprises a set of points $V_t \subseteq V$. Using knowledge of prior instances $V_0, V_1, \ldots, V_{t-1}$, the learning-augmented algorithm has to produce a solution $Y_t$ comprising $k$ points in $V_{<t} := V_0 \cup V_1 \cup \ldots \cup V_{t-1}$.[2] This solution is then applied to the instance $V_t$ and incurs cost

$$\mathsf{Cost}_t(Y_t) := \sum_{x \in V_t} D(Y_t, x), \text{ where } D(Y_t, x) := \min_{y \in Y_t} d(x, y).$$

Correspondingly, the (approximation) loss of the algorithm is given by

$$\rho(Y_t, V_t) = \frac{\mathsf{Cost}_t(Y_t)}{\mathsf{OPT}_t}, \text{ where } \mathsf{OPT}_t := \min_{Y \subseteq V, |Y| = k} \sum_{x \in V_t} D(Y, x).$$

Since the ratio defining $\rho(Y_t, \mathcal{I}_t)$ is invariant under scaling, we may assume that $\min_{x, y \in V; x \neq y} d(x, y) = 1$, and denote the *aspect ratio* $\max_{x, y \in V} d(x, y)$ by $\Delta$. Also, to avoid degenerate cases where $\mathsf{OPT}_t = 0$, causing the ratio to be infinity, we assume each instance must have at least $k + 1$ distinct points.

The goal is to match the average performance over time of the best fixed solution $Y^*$ (in hindsight), where $Y^* := \operatorname{argmin}_{Y \subseteq V, |Y| = k} \sum_{t=1}^{T} \rho(Y, V_t)$.

**Our Results.** We say that a learning-augmented algorithm for LEARN-MEDIAN obtains an $(\alpha, \beta)$-approximation if

$$\sum_{t=1}^{T} \rho(Y_t, V_t) \leq \alpha \cdot \sum_{t=1}^{T} \rho(Y^*, V_t) + \beta.$$

We call $\alpha$ the competitive ratio and $\beta$ the regret of the algorithm. The goal is to obtain sublinear regret, i.e., $\beta = o(T)$, while minimizing the competitive ratio $\alpha$.

Our first result is a *deterministic* algorithm with $\alpha = O(k), \beta = o(T)$. Precisely, we get

$$\sum_{t=1}^{T} \rho(Y_t, V_t) \leq O(k) \cdot \sum_{t=1}^{T} \rho(Y^*, V_t) + O\left(k^4 \Delta \cdot \sqrt{T} \log(T) \log(Tk)\right). \tag{1}$$

We also give a matching lower bound showing that the competitive ratio of $O(k)$ is necessary, i.e., there is no deterministic algorithm with competitive ratio $\alpha = o(k)$ and sub-linear regret $\beta = o(T)$.

Next, we use randomization to improve the competitive ratio to $O(1)$. We give a *randomized* algorithm with $\alpha = O(1), \beta = o(T)$:

$$\sum_{t=1}^{T} \mathbb{E}[\rho(Y_t, V_t)] \leq O(1) \cdot \sum_{t=1}^{T} \rho(Y^*, V_t) + O\left(k^3 \Delta \cdot \sqrt{T} \log(T) \log(Tk)\right). \tag{2}$$

As in the deterministic case, we show that the $O(1)$ factor in the competitive ratio is necessary, i.e., there is no algorithm (randomized or deterministic) with competitive ratio $\alpha = 1 + \varepsilon$ for arbitrarily small $\varepsilon > 0$ and sub-linear regret $\beta = o(T)$.

Finally, note that the regret bounds in both results depend on $k, \Delta$. We show that this dependence is necessary, i.e., there is no algorithm (randomized or deterministic) with competitive ratio $\alpha = O(1)$ and regret $\beta = o(k\Delta)$.

**Our Techniques.** We give a sketch of the main techniques in our algorithms. Both the deterministic and randomized algorithm share all the steps, except the last one.

---

[2] The set $V_0$ initializes the problem. The algorithm does not generate a solution for $V_0$.

- In the first step, we convert the LEARN-MEDIAN instance to a bounded version of this problem where each instance comprises exactly $k$ points in the metric space. We call this latter problem LEARN-BOUNDED-MEDIAN. The conversion is done online at every time $t$.

- Next, we relax the $k$-median problem by allowing fractional solutions, i.e., the $k$ centers are allowed to be distributed fractionally across the points of the metric space. Our main technical work is in producing a fractional solution to the $k$-median instances of LEARN-BOUNDED-MEDIAN. Note that the feasible fractional solution define a (convex) simplex, but the number of dimensions of the simplex grows over time as more points appear in the $k$-median instances. The main algorithmic ideas that we develop for this problem are outlined in the paragraph below.

- Our final step is in rounding the fractional solutions to integer $k$-median solutions. Here, we give two different algorithms, one deterministic and the other randomized, both of which adapt ideas from LP-based rounding of $k$-median.

We now outline the main ideas in obtaining a fractional solution to the $k$-median instances. The main challenge is that the convex space of solutions available to the algorithm expands over time, yet the algorithm must compete with a solution that is only guaranteed to lie in the final convex set. This means that in applying standard online convex optimization (OCO) tools such as online mirror descent (OMD), we only have partial knowledge of the gradient at any intermediate step corresponding to the dimensions/points that have appeared previously. Unfortunately, this limitation makes standard regularizers ineffective: the squared $L_2$-norm regularizer fails to achieve sub-linear regret since the dimension of the gradient depends on the time horizon, while the negative entropy regularizer is unable to handle new dimensions since the gradient is undefined at zero. Instead, we give a non-standard changing regularizer parametrized by the number of unique points seen so far. Correspondingly, we depart from standard OCO analysis that compares against a fixed solution in hindsight. Instead, we divide the timeline into phases and compare the performance of our algorithm against a dynamic set of locally optimal solutions. The main advantage is that within a phase, we only compare our algorithm against a solution that exists in the current convex set. The phases are chosen carefully so that we can use metric properties to show that these locally optimal solutions have total cost comparable to the hindsight optimum. Morally, we ensure that if the hindsight optimum is far from the locally optimal solutions, then it performs poorly on the instances in these phases, and therefore, cannot be the best solution in hindsight across all the instances. Putting all these components together in an online learning algorithm is the main technical contribution of this paper.

**Experimental Evaluation.** We perform experiments to validate our theoretical results empirically. We first run our algorithms on synthetic data generated from simple distributions such as a uniform distribution on a square or on a set of distinct clusters. In these cases, we confirm that the algorithms quickly converge to the optimal solution, and the average performance is nearly optimal in long run, thereby significantly outperforming theoretical bounds. In our second set of experiments, we use a sequence of dynamically changing distributions with the goal of understanding the responsiveness of our algorithm to changing data. We observe that as the algorithm discovers new data sets corresponding to new regions of the metric space, it reacts quickly to the changing environment and changes the location of $k$ centers, in some cases even outperforms the static best solution in hindsight.

## 1.2 Related Work

Our work fits in the burgeoning field of learning-augmented algorithms, where the goal is to leverage ML predictions to improve the performance of algorithms for optimization problems. This paradigm was initially proposed for the online caching problem by [Lykouris and Vassilvitskii, 2021], and has since been applied to many problem domains (see [Mitzenmacher and Vassilvitskii, 2022] for a survey). In particular, various clustering problems such as $k$-median and $k$-means clustering [Ergun et al., 2022, Nguyen et al., 2023, Gamlath et al., 2022] and facility location [Jiang et al., 2022, Fotakis et al., 2025] have been studied in this framework, both in the offline and online settings. Our work sharply deviates from these existing lines of research in that we consider a model comprising a sequence of problem instances inspired by online learning, while prior work focused on solving a single instance using ML predictions. A notable exception is Khodak et al. [2022], where the authors consider an online learning based model to learn *predictions* from a sequence of problem instances, and apply it to a variety of matching, scheduling, and rent-or-buy problems. In some sense, our

model is a more direct counterpart of this idea, in that we apply the online learning paradigm to the performance of the algorithm directly instead of using a surrogate learned parameter.

From a technical standpoint, our work is related to the field of combinatorial online learning, where the goal is to develop low-regret algorithms for problems with exponentially large discrete action spaces. Traditional online learning tools like multiplicative weights update (MWU) are typically inapplicable due to computational barriers stemming from the sheer size of the solution space. To address this, several works have adapted online convex optimization and gradient-based techniques for combinatorial problems. Applications include online facility location [Christou et al., 2023, Pasteris et al., 2021], online $k$-means clustering [Cohen-Addad et al., 2021], online routing [Awerbuch and Kleinberg, 2008], online learning of disjunctions [Helmbold et al., 2002], coalition formation [Cohen and Agmon, 2024], online submodular minimization [Hazan and Kale, 2012], online matrix prediction [Hazan et al., 2012], and online ranking [Fotakis et al., 2020]. One distinction in our model is that we consider the average approximation ratio across the instances, while these works consider average cost. More importantly, the novelty in our work lies in designing a new online learning algorithm and analysis for the fractional problem, while most papers focus on reducing the combinatorial problem to an OCO formulation and apply standard OCO tools to the resulting problem.

Our work also bears some similarity with prior work on facility location problems in evolving metric spaces [Eisenstat et al., 2014, An et al., 2017]. But, in technical terms, this line of research is quite different in that it focuses on metric distances changing over time, while our focus is on new locations being added. Another related line of work is that of universal algorithms for optimization problems. These algorithms aim to compute a single solution that performs well across all possible realizations of the input, thus providing robust approximation guarantees. Examples include universal algorithms for Steiner tree and TSP [Jia et al., 2005, Bhalgat et al., 2011, Gupta et al., 2006], set cover [Grandoni et al., 2008, Jia et al., 2005], clustering [Ganesh et al., 2023]. Unlike our work which is in the online setting, universal algorithms are offline and thereby use a very different set of tools.

Finally, our work is also related to data-driven algorithms (see [Balcan, 2021] for a survey), where the typical goal is to pick the best algorithm out of a palette of algorithms whose behavior is dictated by a numerical parameter. In this line of work, the typical assumption is that the instances are drawn from an unknown distribution, and the goal is to obtain sample complexity bounds for a given target average performance in comparison to the best parameter choice. In contrast, in our setting, the algorithm is allowed to output any valid solution in each round and the final results are about average performance across an adversarial set of instances. [Balcan et al., 2018] considers the parameterized framework but in an online learning setting, where they pick a parameter for each instance that arrives online. The benchmark in this case is the algorithm corresponding to the best fixed parameter in hindsight. Their techniques heavily utilize the parametrization of the algorithms, because of which they are not applicable to our setting.

**Outline of the paper.** In Section 2, we give the reduction from LEARN-MEDIAN to LEARN-BOUNDED-MEDIAN, for which we then give a fractional algorithm in Section 3, and integer algorithms via online rounding in Section 4. Details of these technical sections appear in the appendix. We present our experiments and empirical results in Section 5; again, details are in the appendix. Finally, the lower bounds that complement our algorithmic results are also given in the appendix.

## 2 Reduction from LEARN-MEDIAN to LEARN-BOUNDED-MEDIAN

One difficulty in designing an algorithm for LEARN-MEDIAN is that the $k$-median instances $V_t$ can have arbitrary size, which would mean that the metric space on which we design the algorithm expands in an arbitrary manner in each online step.

To alleviate this problem, in this section, we give a reduction from LEARN-MEDIAN to a simpler problem we call LEARN-BOUNDED-MEDIAN, which has at most $k$ weighted points per instance. In particular, this reduction eliminates the dependence on the size of instances $n$ in our regret bounds. Otherwise, We will see in the next section that the subgradient that we compute at each step of the fractional algorithm can have a large $L_\infty$ norm that depends on $n$, which would then affect the regret term. On the other hand, if we apply our mirror descent algorithm (given in the next section) to only $k$ points in each round, then the total number of points under consideration is at most $kT$. Thus, the regret term is independent of $n$. A second benefit of the reduction is that it reduces the running time of the learning algorithm, and makes it independent of the size of the instances $n$. (The reduction uses a

$k$-median algorithm that runs on instances of size $n$, but this is unavoidable in any case.) We typically expect that $n$ will be much larger than all other parameters in the problem. Using the reduction, we only need to run our learning algorithm on at most $kT$ points, which is independent of $n$.

It is important to choose exactly $k$ points in the reduction because if we use fewer or more than $k$ points, then the weights for the reduced instance are inconsistent with the $k$-median optimum. For example, if we use $< k$ points and there are $k$ closely knit clusters, then the resulting cost of the solution (which determines weights) is arbitrarily large in comparison. Similarly, if we use $> k$ points and there are $k + 1$ closely knit clusters, then the resulting cost of the solution is arbitrarily small.

**Bounded Instances of** LEARN-MEDIAN. Similar to LEARN-MEDIAN, a LEARN-BOUNDED-MEDIAN instance is specified by a metric space $\mathcal{M} = (V, d)$ and we are also given a sequence of $k$-median instances. At time $t$, instance $R_t$ is specified by a set of at most $k$ weighted points. Each point $x \in R_t$ has an associated weight $w_x^t$ (for simplicity, we often omit the superscript $t$ when it is clear from context). The instances further satisfy $\sum_{x \in R_t} w_x \leq (k + 1)$. All other definitions are the same as before, except that the cost is now defined with respect to the weights:

$$\mathsf{Cost}_t(Y, R_t) = \sum_{x \in R_t} w_x D(Y, x).$$

We often just use $\mathsf{Cost}_t(Y)$ when the instance is clear from context.

**Reduction to** LEARN-BOUNDED-MEDIAN. We show the following reduction:

**Theorem 2.1.** *Given an instance for the* LEARN-MEDIAN *problem, there exists an algorithm $\mathcal{A}$ that maps each sub-instance $V_t$ to a sub-instance $R_t = \mathcal{A}(V_t)$, resulting in a* LEARN-BOUNDED-MEDIAN *instance. If solutions $Y_1, ..., Y_T$ for the new instance of* LEARN-BOUNDED-MEDIAN *satisfy*

$$\sum_{t=1}^{T} \mathsf{Cost}_t(Y_t, R_t) \leq \alpha \cdot \min_{\widehat{Y} \subseteq R, |\widehat{Y}| = k} \sum_{t=1}^{T} \mathsf{Cost}_t(\widehat{Y}, R_t) + \gamma,$$

*where $\alpha \geq 1$ and $R = R_0 \cup R_1 \ldots \cup R_T$, then we have*

$$\sum_{t=1}^{T} \rho(Y_t, V_t) \leq O(\alpha) \cdot \min_{Y^* \subseteq V, |Y^*| = k} \sum_{t=1}^{T} \rho(Y^*, V_t) + O(\gamma).$$

We now describe the reduction – the detailed analysis is deferred to the appendix. The main idea is to replace each set $V_t$ by $k$ *centers* approximating the optimal $k$-median cost of $V_t$ – these weighted centers act as proxies for the points in $V_t$. More formally, consider an offline instance of the $k$-median problem comprising the points in $V_t$. We use a constant factor approximation algorithm $\mathcal{A}$ for $\min_{C \subseteq V_t, |C| = k} \sum_{x \in V_t} D(C, x)$ to obtain a set of $k$ centers $C_t = \{c_1, \ldots, c_k\}$. Let $V_t^{(i)}$ denote the points in $V_t$ that are assigned to $c_i$ in the approximate solution. In the weighted instance $R_t$, we choose $R_t = C_t$, and the weight of a point $c_i \in R_t$ is set to $\frac{|V_t^{(i)}|}{\sum_{x \in V_t} D(C_t, x)}$. This completes the reduction to an instance of LEARN-BOUNDED-MEDIAN.

## 3 Fractional Algorithm via Online Mirror Descent with Hyperbolic Entropy Regularizer

To design an online algorithm for LEARN-BOUNDED-MEDIAN, perhaps the simplest idea is *follow the leader* (FTL), where the algorithm outputs the best fixed solution for the prior instances. However, finding the exact FTL solution is computationally intractable; moreover, we show (in the appendix) that using an approximate FTL solution fails to give a competitive algorithm even for $k = 1$.

A common approach to address intractability is to use a convex relaxation. Indeed, [Fotakis et al., 2021] introduced a method that could solve LEARN-BOUNDED-MEDIAN instances if the entire metric space $(V, d)$ were known upfront. Unfortunately, not knowing the metric space changes the problem significantly. For example, if the metric space is known, then it is possible to achieve *pure (sub-linear) regret* using a fractional solution, i.e., a competitive ratio of 1. In contrast, we give a simple example in the appendix that rules this out even for $k = 1$ in our setting.

Our algorithm maintains a fractional solution for points in $R_{<t}$, updated by online mirror descent (OMD) using a hyperbolic entropy regularizer (Definition 3.1). New points introduced have fractional value initialized to 0. To analyze the performance of the algorithm, we construct a sequence of solutions that only uses points in $R_{<t}$ and can only change $k \log T$ times. This allows us to show that the fractional solution is constant competitive with sublinear regret. We give further details of these steps in the rest of this section.

**Fractional solutions for** LEARN-BOUNDED-MEDIAN. At time $t$, only points in $R_{<t}$ are revealed. Let $d_{t-1} = |R_{<t}|$ be the number of points available at time $t$, and $\Delta_{d_t}^k$ denote $\{z \in \mathbb{R}_{\geq 0}^{d_t} : ||z||_1 = k\}$. A fractional solution $y_t$ is in the set $K_{t-1}$ which is the intersection [3] of $\Delta_{d_{t-1}}^k$ and $[0,1]^{d_{t-1}}$.

Given a vector $y_t \in K_{t-1}$, for each point $x$ the fractional assignment cost is defined by assigning the point $x$ fractionally to the centers in $y_t$: $D(y_t, x) := \min_{0 \leq \alpha_i \leq (y_t)_i, \sum_i \alpha_i = 1} \sum_{i \in [d_{t-1}]} \alpha_i d(v_i, x)$. The cost for the instance is still the weighted sum as before: $\mathsf{Cost}_t(y_t) := \sum_{x \in R_t} w_x D(y_t, x)$.

**Our algorithm.** At each time $t$, a weighted subset of points $R_t$ arrives. At time $t = 0$, we initialize $y_0$ (which is a $k$-dimensional vector) to any $k$ arbitrary points in $R_0$. At step $t$, we maintain a fractional solution $y_t$ on the points $R_{<t}$. To go from $y_t$ to $y_{t+1}$, we perform one step of mirror descent based on the $\beta$-hyperbolic entropy regularizer defined below:

**Definition 3.1** ($\beta$-hyperbolic entropy [Ghai et al., 2020]). For any $x \in \mathbb{R}^d$ and $\beta > 0$, define the $\beta$-hyperbolic entropy of $x$, denoted $\phi_\beta(x)$ as:

$$\phi_\beta(x) := \sum_{i=1}^d x_i \operatorname{arcsinh}\left(\frac{x_i}{\beta}\right) - \sqrt{x_i^2 + \beta^2}.$$

The associated Bregman divergence is given by $B_{\phi_\beta}(x||y) = \phi_\beta(x) - \phi_\beta(y) - \langle \nabla \phi_\beta(y), x - y \rangle =$

$$\sum_{i=1}^d \left[ x_i \left( \operatorname{arcsinh}\left(\frac{x_i}{\beta}\right) - \operatorname{arcsinh}\left(\frac{y_i}{\beta}\right) \right) - \sqrt{x_i^2 + \beta^2} + \sqrt{y_i^2 + \beta^2} \right].$$

Note that we considered other regularizers such as $1/2||x||_2^2$ (which recovers online gradient descent) and $\sum_i x_i \log x_i$ (which recovers exponentiated gradient descent). The problem with the former is that the regret term in our analysis is not sublinear in $T$, while the latter is ill-defined for our setting as the gradient is not defined at 0, which we need for the new points. Before taking one mirror descent step, for points in $R_t$ but not in $R_{<t}$, we set their fractional values to 0. This gives the fractional solution $y_{t+1}$ for the next iteration. The detailed algorithm is given in Algorithm 1.

**Analysis.** To prove Algorithm 1 is constant competitive, we compare its cost to a changing sequence of optimal solutions. To construct such a sequence of optimal solutions, we first partition the sequence of $k$-median instances into phases. Let $y^* \in K_T$ be an integral solution defined by a set $C$ of $k$ centers. Let the centers in $C$ be $c^{(1)}, \ldots, c^{(k)}$. The set $C$ partitions $V$ into $k$ subsets – let $V^{(i)}$ be the subset of $V$ for which the closest center is $c^{(i)}$ (we break ties arbitrarily).

Considering each center in this optimal solution separately, we split the solution into different phases at times $j_i$, which are the smallest time $t$ such that the total weight of the points in $V^{(i)} \cap R_{\leq t}$ (i.e., points in $V^{(i)}$ arriving by time $t$) exceeds $k \cdot 2^j$. Define the set $P := \{j_i : i \in [k], j \in [\log(w(V^{(i)}))]\} \subseteq [1, T]$ be the set of times when a new phase starts. It's easy to check that $|P| \leq k \log T$ as the total weight of arriving points is at most $k + 1$ in each step.

For each phase, we bound the cost of the algorithm using standard tools from the OCO literature:

**Lemma 3.2.** *For any time interval $I := [t_a, t_b] \subseteq [1, T]$, let $z$ be an integral vector in $K_{t_a}$. Then,*

$$\sum_{t \in I} (\mathsf{Cost}_t(y_t) - \mathsf{Cost}_t(z)) \leq 3(k+1)k\Delta\sqrt{T}\log(7Tk)$$

This lemma shows that up to additive factors, in each phase we can be competitive against any solution. Therefore it remains to show that there exists a series of solutions that only use available points whose cost is competitive against the offline optimal solution $y^*$.

---

[3]In fact, we take $K_{t-1} = \Delta_{d_{t-1}}^k$ in the experiments for simplicity as the analysis is the same.

**Algorithm 1:** Algorithm for an instance of LEARN-BOUNDED-MEDIAN

---

**1.1** Initialize $y_0$ to any $k$ points in $R_0$, i.e., $y_0 \in \mathbb{R}^k$ with $(y_0)_i = 1 \ \forall i \in [k]$.

**1.2** **for** $t = 1, 2, \ldots, T$ **do**

**1.3**     Let the points in $R_{\leq t-1}$ be $v_1, \ldots, v_{d_{t-1}}$.

**1.4**     **for** *each* $x \in R_t$ **do**

**1.5**         let $\alpha_i^x$ be the fractional assignment of $x$ to $v_i$, i.e., $D(y_t, x) = \sum_i \alpha_i^x d(v_i, x)$.

**1.6**         Define $M^{(x)} := \max_{i : \alpha_i^x > 0} d(v_i, x)$.

**1.7**     **Sub-gradient Step:** Define $\nabla_t \in \mathbb{R}^{d_t}$ as follows: for each $i \in [d_t]$,

$$(\nabla_t)_i := -\sum_{x \in R_t} w_x (M^{(x)} - \min(M^{(x)}, d(x, v_i))).$$

**1.8**     **Learning Rate:** Set

$$\eta_t := \frac{1}{G_t \sqrt{t}}$$

    where $G_t = \max_{t' \leq t} ||\nabla_{t'}||_\infty$.

**1.9**     **Update Step:** Define a vector $x_{t+1} \in \mathbb{R}^{d_t}$ as follows: for each $i \in [d_t]$,

$$(x_{t+1})_i := \frac{\sinh(\operatorname{arcsinh}(d_t(y_t)_i - \eta_t(\nabla_t)_i)}{d_t},$$

    where we use $(y_t)_i = 0$ in the above equation for each $i \in [d_t] \setminus [d_{t-1}]$.

**1.10**     **Projection Step:** Define

$$y_{t+1} := \operatorname{argmin}_{y \in K_t} B_{\phi_{1/d_t}}(y || x_{t+1})$$

---

**Lemma 3.3.** *For each $p \in P$, let the start and end time of phase $p$ be denoted $s_p$ and $e_p$ respectively. Then, there exist solutions $z^{(p)}$ such that*

$$\sum_{p \in P} \sum_{t=s_p}^{e_p} \mathsf{Cost}_t(z^{(p)}) = O\left(k^2 \Delta + \sum_{t=1}^{T} \mathsf{Cost}_t(y^*)\right),$$

*where $y^*$ is an arbitrary integral vector in $K_T$.*

## 4   Integral Solutions via Online Rounding

In this section, we give online rounding algorithms for the LEARN-BOUNDED-MEDIAN problem. Let $y_t$ be the fractional solution computed by Algorithm 1 before the arrival of the $t$th instance $R_t$. Our rounding algorithms will take the fractional solution $y_t$ as input and return an integer solution $Y_t$ with exactly $k$ centers. The *rounding loss* is defined as the ratio of the (expected) cost of the integer solution $Y_t$ to that of the fractional solution $y_t$.

We give two rounding algorithms. The first algorithm is deterministic and has a rounding loss of $O(k)$. The second algorithm is randomized and has only $O(1)$ rounding loss. These are adaptations of existing rounding algorithms for $k$-median [Charikar et al., 1999, Charikar and Li, 2012]; the main difference is that we need to bound the rounding loss for points we haven't seen as well.

**Deterministic Rounding Algorithm.** The deterministic rounding algorithm is based on a natural greedy strategy: at time $t$, maintain a set $Y_t$ that is initially empty. Consider the points in $R_{<t} = R_0 \cup \ldots R_{t-1}$ in non-decreasing order of their connection costs $D(y_t, i)$. When considering a point $i$, add it to $Y_t$ if its connection cost to $Y_t$ exceeds its fractional connection cost in $y_t$ by an $\Omega(k)$ factor. This process ensures that $|Y_t| \leq k$ – indeed, the intuition is that each time we add a point to $Y_t$, we can draw a ball around it which has almost 1 unit of fractional mass (according to $y_t$). The criteria for adding a point to $Y_t$ ensures that the total connection cost to $Y_t$ remains within $O(k)$ factor of that to $y_t$. Thus, we get a rounding algorithm that rounds $y_t$ to an integral solution $Y_t$ and loses an $O(k)$

factor in approximation – we give the details in the appendix. Combining this fact with Theorem 2.1 and Lemma 3.3, we get the desired result (1).

**Randomized Rounding Algorithm.** The randomized rounding algorithm proceeds in two phases: the first phase is similar to the deterministic rounding algorithm described above. However, instead of keeping a threshold of $\Omega(k)$ for adding a center to $Y_t$, it uses a constant threshold. This results in $|Y_t|$ being larger than $k$. Subsequently a randomized strategy is used to prune this set to size $k$. We show in the appendix that this algorithm has constant approximation gap. Combining this fact with Theorem 2.1 and Lemma 3.3, we get the desired result (2).

In conclusion, we obtain that the time complexity of our overall algorithm (besides the reduction step) is $O(k^2T^3)$, as the bottleneck is computing the $O(kT \times kT)$ distance matrix in each round.

## 5  Experimental Results

In this section, we empirically evaluate the performance of our algorithms. In all our experiments, we use a heuristic to improve the performance of the rounding algorithms, where we do a binary search over the best threshold on the distance of centers to open (this value was $(2k + 2)D(y_t, i)$ for deterministic rounding and $4D(y_t, i)$ for randomized rounding). We set this threshold to the smallest possible that still gives the desired number of centers $k$. We report the main experimental findings in this section, and give additional results and experiments in the appendix.

Our first two experiments are for simple i.i.d. instances. In **Uniform Square**, each round has a set of points sampled uniformly from the unit square $[0, 1] \times [0, 1]$. In **Multiple Clusters**, the underlying metric space consists of a set of clusters, and the points are sampled uniformly from these clusters. In both cases, the distances are given by the underlying Euclidean metric. As seen in Fig. 1, both the deterministic and randomized algorithms converge to expected solutions – for **Uniform Square**, the centers are distributed (roughly) uniformly over the square and for **Multiple Clusters**, the centers spread themselves on the different clusters. In the adjoining plots, we show the approximation ratio of the algorithms, which approaches 1 once the influence of additive regret declines.

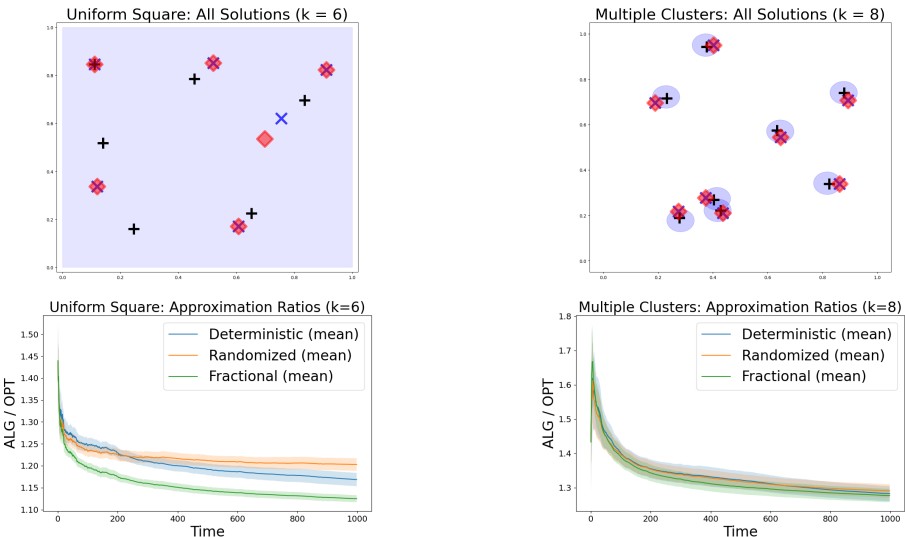

Figure 1: For i.i.d. instances (**Uniform Square** and **Multiple Clusters**), the optimal (black plus), deterministic (blue cross), and randomized (red diamond) solutions (top figure) and approximation ratios - avg. and std. dev. over 10 random instances (bottom figure).

In the third experiment **Oscillating Instances**, we have two distinct clusters, and instances comprise alternating batches of geometrically increasing length that draws points from one of the clusters. The optimal solution alternates between the two clusters, switching every time the current batch becomes long enough to dominate all previous instances. In Fig. 2a, we show two important properties of our algorithm in this setup. First, for $k = 1$, we plot the ratio over time $t$ between $\sum_{\tau=1}^{t} \rho(Y_\tau, V_\tau)$ and

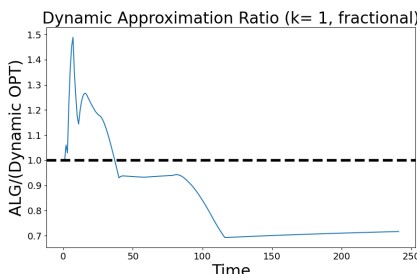

(a) Approximation ratio for **Oscillating Instances**

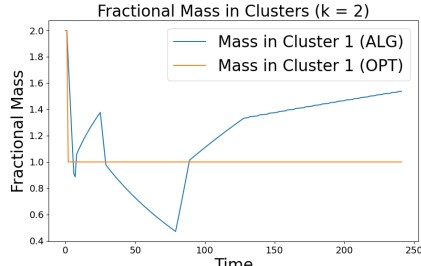

(b) Fractional mass of algorithm and OPT over time in one of the clusters for **Oscillating Instances**

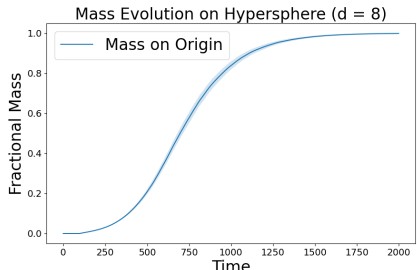

(c) Fractional mass of algorithm over time at the center for **Uniform Hypersphere** (avg. and std. dev. over 10 runs)

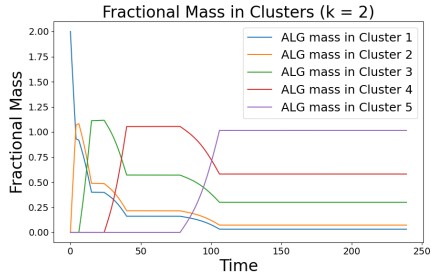

(d) Fractional mass of algorithm in each cluster in **Scale Changing**

Figure 2: Experimental Results for Dynamic Instances

$\sum_{\tau=1}^{t} \rho(\mathsf{OPT}_t, V_\tau)$, where $\mathsf{OPT}_t$ is the best fixed solution for instances $V_1, ..., V_t$. Interestingly, our algorithm outperforms the optimal solution by virtue of being adaptive while the optimal solution is fixed. Next, in Fig. 2b we trace the migration of fractional mass between the two clusters for $k = 2$. We observe the algorithm quickly moves one unit of mass to the current cluster, and then slows down since moving more fractional mass does not significantly reduce cost.

In the next set of experiments, we explore settings where the metric space changes significantly over time. In **Uniform Hypersphere**, we choose $k = 1$ and every round consists of points chosen at random from the surface of the unit sphere. Around the 100th iteration, we also include the origin in the instance. This changes the space of solutions drastically because the origin is now a much better solution than any of the surface points. As seen in Fig. 2c, the algorithm moves fractional mass to the origin once it becomes available, eventually converging entirely to this location.

Our final experiment is called **Scale Changing**. Here, we have 5 clusters with 10 points each. The clusters grow geometrically far apart in distance. The sequence of instances iterates over these clusters in batches of geometrically increasing length. Fig. 2d shows the fractional mass of the algorithm's solution in the all clusters. We see that our algorithm adapts to the changing environment by increasing the fractional mass in the cluster being currently presented, and as earlier, this process slows after unit fractional mass has been moved to the new cluster.

## 6 Closing Remarks

In this paper, we studied the problem of solving a sequence of $k$-median clustering instances to match the best fixed solution in hindsight. One limitation of our techniques is that they are tailored to metric problems. While these techniques might extend to problems similar to $k$-median such as $k$-means clustering and facility location, they do not apply to other problem domains such as covering for which learning-augmented algorithms have been extensively studied. Nevertheless, our problem formulation bridging online learning and discrete optimization applies to the entire range of problems in combinatorial optimization. We leave the design of algorithms in our framework for other problem domains as interesting future work.

## Acknowledgments and Disclosure of Funding

Debmalya Panigrahi and Anish Hebbar were supported in part by NSF awards CCF-195570 and CCF-2329230. Rong Ge was supported in part by NSF Award DMS-2031849.

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

## Technical Appendices and Supplementary Material

## A  Reduction from LEARN-MEDIAN to LEARN-BOUNDED-MEDIAN

Recall that in Section 2 we first reduce LEARN-MEDIAN to a simpler problem we call LEARN-BOUNDED-MEDIAN, which has at most $k$ weighted points per instance. We do this because the original $k$-median instances $V_t$ can have arbitrary size, which would mean that the metric space on which we design the algorithm expands in an arbitrary manner in each online step.

We give the details of this reduction step in this section.

### A.1  Bounded Instances of LEARN-MEDIAN

A LEARN-BOUNDED-MEDIAN instance is specified by a metric space $\mathcal{M} = (V, d)$ and we are also given a sequence of subinstances. At time $t$, instance $R_t$ is specified by a set of at most $k$ weighted points $R_t$. Each point $x \in R_t$ has an associated weight $w_x^t$ (for simplicity, we often omit the superscript $t$ when it is clear from context). The instances further satisfy $\sum_{x \in R_t} w_x \leq (k+1)$. The cost for a round is now given by the weighted $k$-median cost:

$$\mathsf{Cost}_t(Y, R_t) = \sum_{x \in R_t} w_x D(Y, x).$$

We often just use $\mathsf{Cost}_t(Y)$ when the instance is clear from context.

### A.2  Reduction

We now show that any instance of LEARN-MEDIAN can be reduced (in an online manner) to an instance of LEARN-BOUNDED-MEDIAN while preserving the competitive ratio up to a constant factor. More formally, we show

**Theorem 2.1.** *Given an instance for the* LEARN-MEDIAN *problem, there exists an algorithm $\mathcal{A}$ that maps each sub-instance $V_t$ to a sub-instance $R_t = \mathcal{A}(V_t)$, resulting in a* LEARN-BOUNDED-MEDIAN *instance. If solutions $Y_1, ..., Y_T$ for the new instance of* LEARN-BOUNDED-MEDIAN *satisfy*

$$\sum_{t=1}^{T} \mathsf{Cost}_t(Y_t, R_t) \leq \alpha \cdot \min_{\widehat{Y} \subseteq R, |\widehat{Y}| = k} \sum_{t=1}^{T} \mathsf{Cost}_t(\widehat{Y}, R_t) + \gamma,$$

*where $\alpha \geq 1$ and $R = R_0 \cup R_1 \ldots \cup R_T$, then we have*

$$\sum_{t=1}^{T} \rho(Y_t, V_t) \leq O(\alpha) \cdot \min_{Y^* \subseteq V, |Y^*| = k} \sum_{t=1}^{T} \rho(Y^*, V_t) + O(\gamma).$$

We will first describe the reduction. When the sub-instance $V_t$ arrives at time $t$, we shall create a corresponding sub-instance $R_t$ as follows: consider an offline instance of the $k$-median problem where the only points available are $V_t$. We use a constant factor approximation algorithm $\mathcal{A}$ for the $k$-median problem to obtain a set of $k$ centers. More formally, we find a constant factor approximation to the following problem:

$$\min_{C \subseteq V_t, |C| = k} \sum_{x \in V_t} D(C, x).$$

Let $C_t$ be the subset of $k$ centers returned by the above approximation algorithm. Let the points in $C_t$ be $\{c_1, \ldots, c_k\}$ and $V_t^{(i)}$ denote the points in $V_t$ that are assigned to $c_i$. Thus,

$$\sum_{x \in V_t} D(C_t, x) = \sum_{i=1}^{k} \sum_{x \in V_t^{(i)}} d(c_i, x).$$

In the sub-instance $R_t$, the set of points is just $R_t = C_t$, and the weight of a point $c_i \in R_t$ is equal to

$$\frac{|V_t^{(i)}|}{\sum_{x \in V_t} D(C_t, x)}.$$

Note that the sum of weights of the above points is

$$\frac{\sum_{i=1}^{k}|V_t^{(i)}|}{\sum_{x\in V_t}D(C_t,x)}=\frac{|V_t|}{\sum_{x\in V_t}D(C_t,x)}\le\frac{|V_t|}{|V_t|-k}\le k+1,$$

because $|V_t|\ge k+1$, and the minimum distance between any two distinct points in the metric space is 1 due to our scaling. This completes the description of the instance of LEARN-BOUNDED-MEDIAN. We now show that a competitive algorithm for the new instance implies the same for the original LEARN-MEDIAN instance, while losing only a constant factor in the competitive ratio.

To prove this, we will need the following well-known fact about $k$-median.

**Claim A.1.** *Consider a weighted instance of the $k$-median problem where the set of clients is given by a subset $I$ of points in the underlying metric space. Let $C^\star$ be an optimal solution (i.e., optimal set of $k$ centers) to the instance and $\widehat{C}$ be an optimal solution where the set of centers is required to be a subset of $I$. Then,*

$$\sum_{x\in I}w_x D(\widehat{C},x)\le 2\cdot\sum_{x\in I}w_x D(C^\star,x).$$

*where $w_x$ is the weight of the client $x$.*

*Proof.* Suppose $y$ was the optimal $k$-median solution for $I$, with centres $y^{(1)},\dots,y^{(k)}$, partitioning $I$ into the corresponding clusters $S_1,\dots S_k$. Note that $y^{(i)}$ is the optimal 1-median of the cluster $S_i$. For any set of clients $X$, let $w(X)=\sum_{x\in X}w_x$ denote the total weight of the set.

$$2w(S_i)\sum_{x\in S_i}w_x d(y^{(i)},x)=2\sum_{x'\in S_i}w_{x'}\sum_{x\in S_i}w_x d(y^{(i)},x)=\sum_{x'\in S_i}\sum_{x\in S_i}w_{x'}w_x d(y^{(i)},x')+w_{x'}w_x d(y^{(i)},x)$$

$$\ge\sum_{x'\in S_i}\sum_{x\in S_i}w_{x'}w_x d(x',x)\quad\text{(Triangle inequality)}$$

We then divide both sides by $w(S_i)$ to obtain

$$2\sum_{x\in S_i}w_x d(y^{(i)},x)\ge\sum_{x'\in S_i}\frac{w_{x'}}{w(S_i)}\sum_{x\in S_i}w_x d(x',x).$$

Since the right hand side is a weighted combination of $\sum_{x\in S_i}w_x d(x',x)$ for different $x'\in S_i$, this implies that $y'^{(i)}=\text{argmin}_{x'\in S_i}\sum_{x\in S_i}w_x d(x',x)$ satisfies

$$\sum_{x\in S_i}w_x d(y'^{(i)},x)\le 2\sum_{x\in S_i}w_x d(y^{(i)},x).$$

If we denote $y'$ as the collection of the $k$ centres $y'^{(i)}$, by adding up the above inequality for each $i\in[k]$ we obtain the desired result.

$$\sum_{x\in I}w_x(y',x)\le\sum_{i=1}^{k}\sum_{x\in S_i}w_x d(y'^{(i)},x)\le 2\sum_{i=1}^{k}\sum_{x\in S_i}w_x d(y^{(i)},x)=2\sum_{x\in I}w_x D(y,x).\quad\square$$

We are now ready to prove the correctness of the reduction.

*Proof of Theorem 2.1.* For a time $t$, the subset $C_t$ is a constant factor approximation to the $k$-median instance specified by the set of points $V_t$. In fact, we have the additional restriction that $C_t$ is a subset of $V_t$. Now, Claim A.1 implies that if $Y_t^*$ is an optimal solution to this $k$-median instance, then

$$\sum_{x\in V_t}D(C_t,x)=O(1)\cdot\sum_{x\in V_t}D(Y_t^*,x)\tag{3}$$

Let $Y_t = \{y_t^{(1)}, \ldots, y_t^{(k)}\}$ and $C_t = \{c_t^{(1)}, \ldots, c_t^{(k)}\}$. Let $V_t^{(j)}$ be the subset of $V_t$ that is assigned to $c_t^{(j)}$, i.e., the points in $V_t$ for which the closest center in $C_t$ is $c_t^{(j)}$. Now we upper bound the competitive ratio of the solution $Y_t$ in the instance $V_t$:

$$\rho(Y_t, V_t) = \frac{\sum_{x \in R_t} D(Y_t, x)}{\sum_{x \in V_t} D(Y_t^*, x)} \leq O(1) \cdot \frac{\sum_{x \in V_t} D(Y_t, x)}{\sum_{x \in V_t} D(C_t, x)} \quad \text{(from (3))}$$

$$= O(1) \cdot \frac{\sum_{j=1}^{k} \sum_{x \in V_t^{(j)}} d(Y_t, x)}{\sum_{x \in V_t} D(C_t, x)}$$

$$\leq O(1) \cdot \frac{\sum_{j=1}^{k} \sum_{x \in V_t^{(j)}} (d(Y_t, c_t^{(j)}) + d(c_t^{(j)}, x))}{\sum_{x \in V_t} D(C_t, x)} \quad \text{(triangle inequality)}$$

$$= O(1) \cdot \frac{\sum_{j=1}^{k} |V_t^{(j)}| d(Y_t, c_t^{(j)}) + \sum_{x \in V_t} D(C_t, x)}{\sum_{x \in V_t} D(C_t, x)}$$

$$= O(1) \cdot \mathsf{Cost}_t(Y_t, R_t) + O(1) \tag{4}$$

Let $Y^*$ be the optimal subset of $k$ points that minimizes $\sum_{t=1}^{T} \rho(Y^*, V_t)$. We now lower bound $\sum_t \rho(Y^*, V_t)$. We again use the partition of $V_t$ into $V_t^{(j)}, j \in [k]$ and triangle inequality to get:

$$\rho(Y^*, V_t) = \frac{\sum_{x \in V_t} D(Y^*, x)}{\sum_{x \in V_t} D(Y_t^*, x)} \geq \frac{\sum_{x \in V_t} D(Y^*, x)}{\sum_{x \in V_t} D(C_t, x)}$$

$$= \frac{\sum_{j=1}^{k} \sum_{x \in V_t^{(j)}} d(Y^*, x)}{\sum_{x \in V_t} D(C_t, x)}$$

$$\geq \frac{\sum_{j=1}^{k} \sum_{x \in V_t^{(j)}} (d(Y^*, c_t^{(j)}) - d(c_t^{(j)}, x))}{\sum_{x \in V_t} D(C_t, x)} \quad \text{(triangle inequality)}$$

$$= \frac{\sum_{j=1}^{k} |V_t^{(j)}| d(Y^*, c_t^{(j)}) - \sum_{x \in V_t} D(C_t, x)}{\sum_{x \in V_t} D(C_t, x)}$$

$$= \mathsf{Cost}_t(Y^*, R_t) - 1. \tag{5}$$

Consequently, combining (4) and (5) we obtain

$$\sum_{t=1}^{T} \rho(Y_t, V_t) \leq O(1) \cdot \sum_{t=1}^{T} \mathsf{Cost}_t(Y_t, R_t) + O(T) \leq O(\alpha) \cdot \sum_{t=1}^{T} \mathsf{Cost}_t(\widehat{Y}, R_t) + O(T + \gamma)$$

$$\leq O(\alpha) \cdot \sum_{t=1}^{T} \mathsf{Cost}_t(Y^*, R_t) + O(T + \gamma) \quad (Claim\ A.1)$$

$$\leq O(\alpha) \cdot \sum_{t=1}^{T} (\rho(Y^*, V_t) + 1) + O(T + \gamma) = O(\alpha) \cdot \sum_{t=1}^{T} \rho(Y^*, V_t) + O(\gamma).$$

where in the last step we used that $\alpha \geq 1$ and $\rho(Y^*, V_t) \geq 1$. This proves the desired result. $\quad\square$

## B  Detailed Proofs for Fractional Algorithm

In this section, we give a detailed description of the online algorithm described in Section 3 for maintaining a fractional solution to an instance of the LEARN-BOUNDED-MEDIAN problem. We also give the missing proofs in the analysis of this algorithm.

**Problem Setup:** Consider an instance $\mathcal{M} = (V, d)$ of the LEARN-BOUNDED-MEDIAN problem. There are $T + 1$ rounds in total, and in each round $t$, a weighted sub-instance $R_t$ arrives. This sub-instance is specified by a set of $k$ points $R_t$ in a metric space $\mathcal{M}$. Note that the points in the metric

space $\mathcal{M}$ are revealed over time, and hence, till time $t$, only the points in $R_{\leq t} := \cup_{t' \leq t} R_t$ in the metric space have been revealed. Let $d_t$ denote $|R_{\leq t}|$, and $\Delta_{d_t}^k$ denote $\{z \in \mathbb{R}_{\geq 0}^{d_t} : ||z||_1 = k\}$ where each coordinate $i$ refers to a corresponding point in $R_{\leq t}$. By ensuring that a point in $R_t$ corresponds to the same coordinate in $\Delta_{d_{t'}}$ for all $t' \geq t$, we have $\Delta_{d_t} \subseteq \Delta_{d_{t'}}$ for all $t, t'$ where $t' \geq t$. Finally, let $K_t$ denote the intersection of $\Delta_{d_t}^k$ and $[0,1]^{d_t}$. In an integral solution, we would require that each time $t$, the algorithm outputs a subset $Y_t$ of $k$ points in $R_{\leq t-1}$; however, in a fractional solution we relax this requirement as follows – the online algorithm outputs a vector $y_t \in K_{t-1}$ for each time $t$. Given a vector $y_t \in K_{t-1}$, the fractional assignment cost at time $t$ is given by:

$$\mathsf{Cost}_t(y_t) := \sum_{x \in R_t} w_x D(y_t, x),$$

where $D(y_t, x)$ is obtained by fractionally assigning $x$ to an extent of $1$ to the closest points in $y_t$. In other words, let $v_i$ denote the point corresponding to coordinate $i$ in $K_{t-1}$. Then, $D(y_t, x) := \min_{\alpha \in [0,1]^{d_{t-1}}} \sum_{i \in [d_{t-1}]} \alpha_i d(v_i, x)$, where $\sum_i \alpha_i = 1$ and $0 \leq \alpha_i \leq (y_t)_i$.

## B.1 Basic facts about the 1-median problem

In this section, we state some basic facts about the $1$-median problem. Given a weighted set $X$ of points in a metric space $\mathcal{M}$, let $\mathsf{opt}_1(X)$ be the optimal weighted $1$-median cost of $X$, i.e.,

$$\mathsf{opt}_1(X) := \min_{y \in \mathcal{M}} \sum_{x \in X} w_x d(y, x).$$

where $w_x$ is the weight of the point $x$. The minimizer $y$ above shall be referred as the *optimal 1-median center* of $X$. The following result shows that $\mathsf{opt}_1(X)$ is closely approximated by a weighted sum of the pair-wise distances between the points in $X$.

**Fact B.1.** *Let $X$ be a weighted set of points in a metric space. Then,*

$$\mathsf{opt}_1(X) \geq \frac{1}{2} \sum_{x \in X} \frac{w_x}{w(X)} \sum_{x' \in X} w_{x'} d(x, x').$$

*Proof.* Let $y^*$ be an optimal $1$-median center of $X$. Then,

$$2w(X) \sum_{x \in X} w_x d(y^*, x) = 2 \sum_{x' \in X} w_{x'} \sum_{x \in X} w_x d(y^*, x) = \sum_{x' \in X} \sum_{x \in X} w_{x'} w_x d(y^*, x) + w_{x'} w_x d(y^*, x')$$

$$\geq \sum_{x' \in X} \sum_{x \in X} w_{x'} w_x d(x, x') \quad \text{(Triangle inequality)}$$

Dividing both sides by $2w(X)$ now gives the desired bound. $\square$

The following result states that the weighted distance of $X$ from a point $v$ can be well-approximated by the weighted distance between $v$ and an optimal $1$-median of $X$.

**Fact B.2.** *Let $X$ be a weighted set of points in a metric space and $y^*$ be an optimal $1$-median of $X$. For any point $v$ in the metric space,*

$$d(v, y^*) \leq 2 \sum_{x \in X} \frac{w_x}{w(X)} d(v, x).$$

*Proof.* We have:

$$\frac{w(X)d(v, y^*)}{w(X)} = \frac{\sum_{x \in X} w_x d(v, y^*)}{w(X)} \leq \frac{\sum_{x \in X} w_x d(v, x)}{w(X)} + \frac{\sum_{x \in X} w_x d(y^*, x)}{w(X)} \quad \text{(Triangle inequality)}$$

$$\leq 2 \sum_{x \in X} \frac{w_x d(v, x)}{w(X)} \quad \text{(Optimality of } y^*\text{)}. \qquad \square$$

Computing the optimal $1$-median center $y^*$ can be hard as we don't know the entire metric space– instead the following result shows that replacing $y^*$ by the optimal center amongst $X$ retains the property stated above.

**Corollary B.3.** *Let $X$ be a set of points in a metric space $\mathcal{M}$ and $v$ be an arbitrary point in $\mathcal{M}$. Let $x^\star \in X$ be a point that minimizes $\sum_{x \in X} w_x d(x^*, x)$. Then,*

$$d(v, x^*) \leq 3 \sum_{x \in X} \frac{w_x}{w(X)} d(v, x)$$

*Proof.* Let $y^*$ be an optimal 1-center of $X$. Using Fact B.1, we have:

$$\sum_{x \in X} w_x d(y^*, x) \geq \frac{1}{2} \min_{x \in X} \sum_{x' \in X} w_{x'} d(x, x') = \frac{1}{2} \sum_{x \in X} w_x d(x^*, x)$$

Proceeding as in the proof of Fact B.2, we get

$$d(v, x^*) = \frac{\sum_{x \in X} w_x d(v, x^*)}{w(X)} \leq \frac{\sum_{x \in X} w_x d(v, x)}{w(X)} + \frac{\sum_{x \in X} w_x d(x^*, x)}{w(X)}.$$

The result now follows from Claim A.1,

$$\sum_{x \in X} w_x d(x^*, x) \leq 2 \sum_{x \in X} w_x d(y^*, x) \leq 2 \sum_{x \in X} w_x d(v, x). \qquad \square$$

## B.2 Regularizer and its properties

In this section, we define the regularizer that our online mirror descent algorithm shall use. We also give useful properties of the regularizer.

**Definition B.4** ($\beta$-hyperbolic entropy). [Ghai et al., 2020] For any $x \in \mathbb{R}^d$ and $\beta > 0$, define the $\beta$-hyperbolic entropy of $x$, denoted $\phi_\beta(x)$ as:

$$\phi_\beta(x) := \sum_{i=1}^{d} x_i \operatorname{arcsinh}\left(\frac{x_i}{\beta}\right) - \sqrt{x_i^2 + \beta^2}.$$

Note that $\phi_\beta(x)$ is convex and twice differentiable, and its gradient is given by

$$(\nabla \phi_\beta(x))_i = \operatorname{arcsinh}(x_i/\beta) \quad \forall i \in [d] \tag{6}$$

Consequently its Hessian is the diagonal matrix,

$$\nabla^2 \phi_\beta(x) = \operatorname{Diag}\left(\frac{1}{x_1^2 + \beta^2}, \frac{1}{x_2^2 + \beta^2}, \cdots, \frac{1}{x_d^2 + \beta^2}\right)$$

We now establish strong convexity properties of the function $\phi_\beta$.

**Definition B.5** (Strongly convex function). A twice differentiable function $f : K \to \mathbb{R}$ on a convex set $K \subseteq \mathbb{R}^d$ is said to be $\alpha$-strongly convex with respect to a norm $|| \cdot ||$ on $K$ if for all $x, y \in K$

$$f(x) - f(y) - \langle \nabla f(y), x - y \rangle \geq \frac{\alpha}{2} ||x - y||^2,$$

or equivalently,

$$\inf_{x \in K, y \in \mathbb{R}^d : ||y|| = 1} y^T \nabla^2 f(x) y \geq \alpha.$$

**Lemma B.6.** *The function $\phi_\beta$ is $\frac{1}{k+\beta d}$-strongly convex over $\Delta_d^k$ with respect to the $\ell_1$ norm.*

*Proof.* We use the second characterization in Definition B.5. Consider vectors $x \in \Delta_d^k$ and $y \in \mathbb{R}^d$ such that $||y||_1 = 1$. Then

$$y^T \nabla^2 \phi_\beta(x) y = \sum_{i=1}^{d} \frac{y_i^2}{\sqrt{\beta^2 + x_i^2}} = \frac{1}{\sum_{i=1}^{d} \sqrt{\beta^2 + x_i^2}} \left(\sum_{i=1}^{d} \frac{y_i^2}{\sqrt{\beta^2 + x_i^2}}\right) \left(\sum_{i=1}^{d} \sqrt{\beta^2 + x_i^2}\right)$$

$$\geq \frac{1}{\sum_{i=1}^{d} \sqrt{\beta^2 + x_i^2}} \left(\sum_{i=1}^{d} |y_i|\right)^2 = \frac{1}{\sum_{i=1}^{d} \sqrt{\beta^2 + x_i^2}} \geq \frac{1}{\sum_{i=1}^{d} (\beta + x_i)} = \frac{1}{k + \beta d},$$

where the first inequality follows from Cauchy-Schwarz. $\qquad \square$

**Corollary B.7.** *The function $\phi_{\frac{1}{d}}$ is $\frac{1}{k+1}$-strongly convex over $\Delta_d^k$ with respect to the $\ell_1$ norm.*

We now recall the notion of Bregman divergence and state some well-known properties [Hazan, 2016].

**Definition B.8** (Bregman Divergence). Let $g : K \to \mathbb{R}$ be a convex function defined on a convex set $K \subseteq \mathbb{R}^d$. Given two points $x, y \in K$, The Bregman divergence $B_g(x||y)$ with respect the function $g$ is defined as

$$B_g(x||y) := g(x) - g(y) - \langle \nabla g(y), x - y \rangle$$

Note that convexity of $g$ implies that $B_g(x||y) \geq 0$.

**Fact B.9** (Law of Cosines for Bregman Divergence). *Let $g : K \to \mathbb{R}$ be a convex differentiable function. Then*

$$\langle \nabla g(y) - \nabla g(z), y - x \rangle = B_g(x||y) + B_g(y||z) - B_g(x||z).$$

**Fact B.10** (Generalized Pythagorean Theorem for Bregman Divergences). *Let $g : K \to \mathbb{R}$ be a convex, differentiable function and $K' \subseteq K$ be a convex subset. Given $x \in K, z \in K'$, let $y \in K'$ be a minimizer of $B_g(y||z)$. Then*

$$B_g(x||y) + B_g(y||z) \leq B_g(x||z).$$

A direct application of the definition of Bregman divergence and (6) yields:

**Claim B.11.** *Given $x, y \in \mathbb{R}^d$,*

$$B_{\phi_\beta}(x||y) = \sum_{i=1}^{d} \left[ x_i \left( \operatorname{arcsinh}\left(\frac{x_i}{\beta}\right) - \operatorname{arcsinh}\left(\frac{y_i}{\beta}\right) \right) - \sqrt{x_i^2 + \beta^2} + \sqrt{y_i^2 + \beta^2} \right].$$

We now give an upper bound on $B_{\phi_\beta}(x||y)$.

**Lemma B.12.** *Let $x, y \in \Delta_d^k$, assume $\beta < 1$ and $||x||_\infty \leq 1$. Then*

$$B_{\phi_\beta}(x||y) \leq k \log(7/\beta).$$

*Proof.* Consider vectors $x$ and $y$ as in the statement of the Lemma. Using Claim B.11, we get

$$
\begin{aligned}
B_{\phi_\beta}(x||y) &= \sum_{i=1}^{d} \left[ x_i \left( \operatorname{arcsinh}\left(\frac{x_i}{\beta}\right) - \operatorname{arcsinh}\left(\frac{y_i}{\beta}\right) \right) - \sqrt{x_i^2 + \beta^2} + \sqrt{y_i^2 + \beta^2} \right] \\
&\leq \sum_{i=1}^{d} \left[ x_i \operatorname{arcsinh}\left(\frac{x_i}{\beta}\right) - \beta + (y_i + \beta) \right] \\
&= k + \sum_{i=1}^{d} x_i \log\left( \frac{x_i + \sqrt{x_i^2 + \beta^2}}{\beta} \right) \leq k + \sum_{i=1}^{d} x_i \log\left( \frac{1 + \sqrt{2}}{\beta} \right) \\
&\leq k + k \log\left( \frac{1 + \sqrt{2}}{\beta} \right) \leq k \log\left( \frac{7}{\beta} \right),
\end{aligned}
$$

where we have used the facts that $x_i \leq 1$ and $\beta < 1$. $\qquad\square$

## B.3 Algorithm

Let us now recall the fractional online algorithm that we described in Section 3. At each time $t$, a weighted subset of points $R_t$ arrives. At time $t = 0$, we initialize $y_0$ (which is a $k$-dimensional vector) to any $k$ arbitrary points in $R_0$. At step $t$, we maintain a fractional solution $y_t$ on the points $R_{<t}$. To go from $y_t$ to $y_{t+1}$, we perform one step of mirror descent based on the $\beta$-hyperbolic entropy regularizer defined earlier. Before taking the mirror descent step, for points in $R_t$ but not in $R_{<t}$, we set their fractional values to 0. This gives the fractional solution $y_{t+1}$ for the next iteration.

---

**Algorithm 1:** Algorithm for an instance of Learn-Bounded-Median

---

**1.1** Initialize $y_0$ to any $k$ points in $R_0$, i.e., $y_0 \in \mathbb{R}^k$ with $(y_0)_i = 1 \quad \forall i \in [k]$.

**1.2** **for** $t = 1, 2, \ldots, T$ **do**

**1.3**     Let the points in $R_{\leq t-1}$ be $v_1, \ldots, v_{d_{t-1}}$.

**1.4**     **for** *each* $x \in R_t$ **do**

**1.5**        let $\alpha_i^x$ be the fractional assignment of $x$ to $v_i$, i.e., $D(y_t, x) = \sum_i \alpha_i^x d(v_i, x)$.

**1.6**        Define $M^{(x)} := \max_{i:\alpha_i^x > 0} d(v_i, x)$.

**1.7**     **Sub-gradient Step:** Define $\nabla_t \in \mathbb{R}^{d_t}$ as follows: for each $i \in [d_t]$,

$$(\nabla_t)_i := -\sum_{x \in R_t} w_x (M^{(x)} - \min(M^{(x)}, d(x, v_i))).$$

**1.8**     **Learning Rate:** Set

$$\eta_t := \frac{1}{G_t \sqrt{t}}$$

where $G_t = \max_{t' \leq t} ||\nabla_{t'}||_\infty$.

**1.9**     **Update Step:** Define a vector $x_{t+1} \in \mathbb{R}^{d_t}$ as follows: for each $i \in [d_t]$,

$$(x_{t+1})_i := \frac{\sinh(\operatorname{arcsinh}(d_t(y_t)_i - \eta_t(\nabla_t)_i)}{d_t},$$

where we use $(y_t)_i = 0$ in the above equation for each $i \in [d_t] \setminus [d_{t-1}]$.

**1.10**     **Projection Step:** Define

$$y_{t+1} := \operatorname{argmin}_{y \in K_t} B_{\phi_{1/d_t}}(y || x_{t+1})$$

---

## B.4   Analysis

In this section, we analyze the algorithm. Recall that the vector $y_t$ lies in $\mathbb{R}^{d_{t-1}}$. But we shall often consider it to lie in $\mathbb{R}^{d_t}$ as well by setting $(y_t)_i$ to 0 for all $i \in [d_t] \setminus [d_{t-1}]$.

**Claim B.13.** *For each $t \in [T]$,*

$$\nabla_t = \frac{1}{\eta_t}\left(\nabla\phi_{1/d_t}(y_t) - \nabla\phi_{1/d_t}(x_{t+1})\right).$$

*Proof.* Using (6) and the definition of $x_{t+1}$, we have that for any $i \in [d_t]$,

$$(\nabla\phi_{1/d_t}(y_t))_i - (\nabla\phi_{1/d_t}(x_{t+1}))_i = \operatorname{arcsinh}(d_t(y_t)_i) - \operatorname{arcsinh}(d_t(x_{t+1})_i)$$
$$= \operatorname{arcsinh}(d_t(y_t)_i - (\operatorname{arcsinh}(d_t(y_t)_i - \eta_t(\nabla_t)_i) = \eta_t(\nabla_t)_i. \quad \square$$

Using the above claim, we bound the difference between $B_{\phi_{1/d_t}}(y_t || x_{t+1})$ and $B_{\phi_{1/d_t}}(y_{t+1} || x_{t+1})$.

**Claim B.14.** *For any time $t$,*

$$B_{\phi_{1/d_t}}(y_t || x_{t+1}) - B_{\phi_{1/d_t}}(y_{t+1} || x_{t+1}) \leq k\eta_t^2 ||\nabla_t||_\infty^2.$$

*Proof.* Using the using the definition of Bregman Divergence, we see that:

$$B_{\phi_{1/d_t}}(y_t||x_{t+1}) - B_{\phi_{1/d_t}}(y_{t+1}||x_{t+1})$$

$$= \phi_{1/d_t}(y_t) - \phi_{1/d_t}(x_{t+1}) - \phi_{1/d_t}(y_{t+1}) + \phi_{1/d_t}(x_{t+1}) - \langle \nabla\phi_{1/d_t}(x_{t+1}), y_t - x_{t+1} + x_{t+1} - y_{t+1} \rangle$$

$$= \phi_{1/d_t}(y_t) - \phi_{1/d_t}(y_{t+1}) - \langle \nabla\phi_{1/d_t}(x_{t+1}), y_t - y_{t+1} \rangle$$

$$= \phi_{1/d_t}(y_t) - \phi_{1/d_t}(y_{t+1}) - \langle \nabla\phi_{1/d_t}(y_t), y_t - y_{t+1} \rangle + \langle \nabla\phi_{1/d_t}(y_t), y_t - y_{t+1} \rangle - \langle \nabla\phi_{1/d_t}(x_{t+1}), y_t - y_{t+1} \rangle$$

$$\overset{\text{Cor. B.7}}{\leq} -\frac{1}{2(k+1)}||y_t - y_{t+1}||_1^2 + \langle \nabla\phi_{1/d_t}(y_t) - \nabla\phi_{1/d_t}(x_{t+1}), y_t - y_{t+1} \rangle$$

$$\overset{\text{Cl. B.13}}{=} -\frac{1}{2(k+1)}||y_t - y_{t+1}||_1^2 + \eta_t \langle \nabla_t, y_t - y_{t+1} \rangle$$

$$\overset{\text{Holder's ineq.}}{\leq} -\frac{1}{2(k+1)}||y_t - y_{t+1}||_1^2 + \eta_t ||\nabla_t||_\infty ||y_t - y_{t+1}||_1.$$

The desired result now follows from the fact that the maximum value of the function $f(z) := -\frac{1}{2(k+1)}z^2 + \eta_t ||\nabla_t||_\infty z$ is at most $k\eta_t^2 ||\nabla_t||_\infty^2$. $\qquad\square$

We now analyze the performance of our algorithm with respect to an arbitrary integral solution.

**Claim B.15.** *Let $y \in K_t$ be an integral vector. Then,*

$$\mathsf{Cost}_t(y) \geq \mathsf{Cost}_t(y_t) + \langle \nabla_t, y - y_t \rangle.$$

*Proof.* Using the definition of $\nabla_t$, it suffices to show that for each $x \in R_t$:

$$D(x, y) \geq D(x, y_t) - \sum_i (M^{(x)} - \min(M^{(x)}, d(x, v_i)))(y_i - (y_t)_i), \qquad (7)$$

where $M^{(x)}$ and $v_i$ are as defined in Algorithm 1. For ease of notation, relabel the points in $R_{\leq t}$ arranged in increasing order of distance from $x$, i.e., $d(x, v_1) \leq d(x, v_2) \leq \ldots \leq d(x, v_{d_t})$. Let $i^*$ be the smallest index such that $\sum_{i \leq i^*}(y_t)_i \geq 1$. Observe that $M^{(x)} = d(x, v_{i^*})$ and $\alpha_i^x = (y_t)_i$ for all $i < i^*$ and is 0 for all $i > i^*$. Since $y$ is an integral vector $D(y, x) = d(v_{i_0}, x)$ for some $i_0 \in [d_t]$. Thus, (7) is equivalent to showing:

$$d(x, v_{i_0}) \geq \sum_{i \leq i^*} \alpha_i^x d(x, v_i) - \sum_{i < i^*}(d(x, v_{i^*}) - d(x, v_i))(y_i - (y_t)_i)$$

$$= \sum_{i < i^*}(y_t)_i d(x, v_i) + \alpha_{i^*}^x d(x, v_{i^*}) - \sum_{i < i^*}(d(x, v_{i^*}) - d(x, v_i))(y_i - (y_t)_i)$$

$$= \alpha_{i^*}^x d(x, v_{i^*}) + d(x, v_{i^*})\sum_{i < i^*}(y_t)_i - \sum_{i < i^*}(d(x, v_{i^*}) - d(x, v_i))y_i$$

$$= d(x, v_{i^*}) - \sum_{i < i^*}(d(x, v_{i^*}) - d(x, v_i))y_i \qquad (8)$$

Two cases arise: (i) $i_0 \geq i^*$, or (ii) $i_0 < i^*$. In the first case, $y_i = 0$ for all $i < i^*$ and hence, (8) follows easily because $d(x, v_{i_0}) \geq d(x, v_{i^*})$. In the second case, the r.h.s. of (8) can be expressed as

$$d(x, v_{i^*}) - (d(x, v_{i^*}) - d(x, v_{i_0})) - \sum_{i_0 < i < i^*}(d(x, v_{i^*}) - d(x, v_i))y_i$$

$$= d(x, v_{i_0}) - \sum_{i_0 < i < i^*}(d(x, v_{i^*}) - d(x, v_i))y_i \leq d(x, v_{i_0}).$$

This proves the desired result. $\qquad\square$

**Phases.** We shall divide the timeline into phases, and shall bound the cost incurred by the algorithm in a phase with respect to an off-line solution that only uses points revealed till the beginning of the phase. The following result bounds the cost incurred during a time interval:

**Lemma 3.2.** *Consider a time interval* $I := [t_a, t_b] \subseteq [1, T]$, *and let* $z$ *be an integral vector in* $K_{t_a}$. *Then,*

$$\sum_{t \in I} (\mathsf{Cost}_t(y_t) - \mathsf{Cost}_t(z)) \leq 3(k+1)k\Delta\sqrt{T}\log(7Tk).$$

*Proof.* Consider a time $t \in I$. Using Claim B.15, we get:

$$\mathsf{Cost}_t(y_t) - \mathsf{Cost}_t(z) \leq \langle \nabla_t, y_t - z \rangle.$$

Summing this over all $t \in I$, and using Claim B.13, we see that $\sum_{t \in I}(\mathsf{Cost}_t(y_t) - \mathsf{Cost}_t(z))$ is at most

$$\sum_{t \in I} \frac{1}{\eta_t} \langle \nabla\phi_{1/d_t}(y_t) - \nabla\phi_{1/d_t}(x_{t+1}), y_t - z \rangle$$

$$\stackrel{\text{Fact B.9}}{=} \sum_{t \in I} \frac{1}{\eta_t} \left( B_{\phi_{1/d_t}}(z||y_t) + B_{\phi_{1/d_t}}(y_t||x_{t+1}) - B_{\phi_{1/d_t}}(z||x_{t+1}) \right)$$

$$\stackrel{\text{Fact B.10}}{=} \sum_{t \in I} \frac{1}{\eta_t} (B_{\phi_{1/d_t}}(z||y_t) + B_{\phi_{1/d_t}}(y_t||x_{t+1}) - B_{\phi_{1/d_t}}(z||y_{t+1}) - B_{\phi_{1/d_t}}(y_{t+1}||x_{t+1}))$$

$$\leq \frac{1}{\eta_{t_a}} B_{\phi_{1/d_{t_a}}}(z||y_{t_a}) + \sum_{t=t_a+1}^{t_b} B_{\phi_{1/d_t}}(z||y_t)\left(\frac{1}{\eta_t} - \frac{1}{\eta_{t-1}}\right) +$$

$$+ \sum_{t \in I} \frac{1}{\eta_t}\left(B_{\phi_{1/d_t}}(y_t||x_{t+1}) - B_{\phi_{1/d_t}}(y_{t+1}||x_{t+1})\right).$$

Using Lemma B.12 and Claim B.14, the above expression is at most (note that $d_t$ is a non-decreasing function of $t$)

$$\frac{k\log(7d_{t_a})}{\eta_{t_a}} + k\log(7d_{t_b}) \cdot \sum_{t=t_a+1}^{t_b}\left(\frac{1}{\eta_t} - \frac{1}{\eta_{t-1}}\right) + \sum_{t \in I} k\eta_t||\nabla_t||_\infty^2$$

$$\leq \frac{k\log(7d_{t_b})}{\eta_{t_b}} + \sum_{t \in I} k\eta_t||\nabla_t||_\infty^2. \tag{9}$$

Observe that $||\nabla_t||_\infty \leq (k+1)\Delta$, and thus $G_t \leq (k+1)\Delta$. Indeed, by definition of $\nabla_t$,

$$|(\nabla_t)_i| \leq \sum_{x \in R_t} w_x|M^{(x)} - d(x, v_i)| \leq \sum_{x \in R_t} w_x\Delta \leq (k+1)\Delta, \tag{10}$$

because the total weight of the points in $R_t$ is at most $(k+1)$. Using this observation and the fact that $\eta_t = \frac{1}{G_t\sqrt{t}}$, (9) is at most

$$(k+1)k\Delta\sqrt{T}\log(7d_T) + \sum_{t \in I}\frac{(k+1)k\Delta}{\sqrt{t}}.$$

The desired result now follows from the fact that $d_T \leq Tk$. $\qquad\square$

We now define the phases. Let $y^* \in K_T$ be an integral solution defined by a set $C$ of $k$ centers. Let $V := R_1 \cup R_2 \ldots \cup R_T$ denote the set of points that arrive over the last $T$ timesteps. Let the centers in $C$ be $c^{(1)}, \ldots, c^{(k)}$. The set $C$ partitions $V$ into $k$ subsets – let $V^{(i)}$ be the subset of $V$ for which the closest center is $c^{(i)}$ (we break ties arbitrarily). For a non-negative integer $j$, let $j_i$ be the smallest time $t$ such that the total weight of the points in $V^{(i)} \cap (R_1 \cup \ldots R_t)$ (i.e., points in $V^{(i)}$ arriving by time $t$) exceeds $k \cdot 2^j$. Let $w(V^{(i)})$ denote the total weight of the points in $V^{(i)}$. Define the set $P := \{j_i : i \in [k], j \in [\log(w(V^{(i)}))]\} \subseteq [1, T]$. Observe that

$$|P| = \sum_{i \in [k]} \log(w(V^{(i)})/k) \leq k\log T, \tag{11}$$

because at each time $t$, the total weight of the arriving points is at most $(k+1)$. We shall treat the indices in $P$ as subset of $[1, T]$, and hence, these partition the timeline into $|P| + 1$ phases. We shall apply Lemma 3.2 for each of these phases.

In order to apply this lemma, we need to specify a candidate solution $z_p$ for each of these phases $p$. We need some more definitions to describe this integral solution. For each $i \in [k]$ and index $j$, define $V_j^{(i)} := V^{(i)} \cap (R_{\leq j_i} \setminus R_{\leq j_{i-1}})$, i.e., the points in $V^{(i)}$ whose corresponding arrival time lies in the range $(j_{i-1}, j_i]$. We shall often refer to the subset $V_j^{(i)}$ as a *bucket* of $V^{(i)}$.

For each of the set of points $V_j^{(i)}$, let $x_j^{(i)}$ denote the optimal 1-median solution where the center is restricted to lie in $V_j^{(i)}$ only, i.e.,

$$x_j^{(i)} := \mathrm{argmin}_{x \in V_j^{(i)}} \, w_x \sum_{v \in V_j^{(i)}} d(x, v).$$

For a phase $p$ and index $i \in [k]$, the time slots in $p$ are contained in a single bucket $V_j^{(i)}$ of $V^{(i)}$. Let $\ell_p^{(i)}$ denote the index corresponding to $x_{j-1}^{(i)}$ in $K_T$, i.e., the optimal 1-median solution of the last bucket in $V^{(i)}$ that does not intersect this phase. We are now ready to define the candidate integral solution $z^{(p)}$ for a phase $p$.

Fix a phase $p$ and let $s_p$ denote the start time of this phase. Define an integral vector $z^{(p)} \in K_{s_p}$ as follows: for each $i \in [k]$, we set $z_j^{(p)} = 1$ where $j = \ell_p^{(i)}$; all other coordinates of $z^{(p)}$ are 0 – in case there is no bucket in $V^{(i)}$ ending before time $s_p$, we set $z^{(p)}$ to be an arbitrary point. In other words, this solution selects $k$ points, namely, the 1-median center of the last bucket in $V^{(i)}$ that ends before $s_p$. This completes the description of the vector $z^{(p)}$ corresponding to a phase $p$. Lemma 3.2 shows that the total cost incurred by the algorithm during a phase $p$ is close to that incurred by the fixed solution $z^{(p)}$. Thus, it remains to show that the total cost incurred by $z^{(p)}$ during a phase is close to that incurred by $y^*$.

**Lemma B.16.** *Consider any $i \in [k]$ and index $j \geq 1$. Then,*

$$\sum_{x \in V_j^{(i)}} w_x d(x_{j-1}^{(i)}, x) \leq 42 \cdot \mathsf{opt}_1(V_{j-1}^{(i)} \cup V_j^{(i)}).$$

*Proof.* Consider a point $x \in V_j^{(i)}$. Using Corollary B.3, we obtain

$$d(x_{j-1}^{(i)}, x) \leq 3 \sum_{x' \in V_{j-1}^{(i)}} \frac{w_{x'}}{w(V_{j-1}^{(i)})} d(x, x') \leq 3 \sum_{x' \in V_{j-1}^{(i)}} \frac{w_{x'}}{k \cdot 2^{j-1}} d(x, x'),$$

where the last inequality follows from the fact that $w(V_{j-1}^{(i)}) \geq k \cdot 2^{j-1}$. Thus,

$$\sum_{x \in V_j^{(i)}} w_x \, d(x_{j-1}^{(i)}, x) \leq 3 \sum_{x \in V_j^{(i)}} \sum_{x' \in V_{j-1}^{(i)}} \frac{w_x w_{x'}}{k \cdot 2^{j-1}} d(x, x').$$

Since the weight of the points arriving at any particular time is at most $k+1$, we have the bound $w(V_{j-1}^{(i)} \cup V_j^{(i)}) \leq k \cdot 2^{j-1} + k \cdot 2^j + 2k + 2 \leq 7k \cdot 2^{j-1}$. Consequently, the r.h.s. above is at most

$$21 \sum_{x \in V_j^{(i)}} \sum_{x' \in V_{j-1}^{(i)}} \frac{w_x w_{x'}}{w(V_{j-1}^{(i)} \cup V_j^{(i)})} d(x, x') \leq 21 \sum_{x, x' \in V_{j-1}^{(i)} \cup V_j^{(i)}} \sum_{x' \in V_{j-1}^{(i)}} \frac{w_x w_{x'}}{w(V_{j-1}^{(i)} \cup V_j^{(i)})} d(x, x'),$$

which is at most $42 \cdot \mathsf{opt}_1(V_{j-1}^{(i)} \cup V_j^{(i)})$ (using Fact B.1). $\qquad\square$

We are now ready to bound the total cost incurred by the solution $z^{(p)}$ during phase $p$.

**Lemma 3.3.** *For each $p \in P$, let the start and end time of phase $p$ be denoted $s_p$ and $e_p$ respectively. Then, there exist solutions $z^{(p)}$ such that*

$$\sum_{p \in P} \sum_{t=s_p}^{e_p} \mathsf{Cost}_t(z^{(p)}) = O\left(k^2 \Delta + \sum_{t=1}^{T} \mathsf{Cost}_t(y^*)\right),$$

*where $y^*$ is an arbitrary integral vector in $K_T$.*

*Proof.* For a point $x \in V^{(i)}$, let $p_x$ be the phase containing the arrival time of $x$. Then,

$$\sum_{p \in P} \sum_{t=s_p}^{e_p} \mathsf{Cost}_t(z^{(p)}) = \sum_{i \in [k]} \sum_{x \in V^{(i)}} D(z^{(p)}, x) \leq \sum_{i \in [k]} \sum_{x \in V^{(i)}} w_x d(x, x_{\ell_p^{(i)}}^{(i)}). \qquad (12)$$

For $x \in V_0^{(i)}$, $d(x_{\ell_p^{(i)}}^{(i)}, x) \leq \Delta$. For $j \geq 1$ and $x \in V_j^{(i)}$, $d(x_{\ell_p^{(i)}}^{(i)}, x) = d(x_{j-1}^{(i)}, x)$. Thus, the r.h.s. of (12) is upper bounded by

$$\sum_{i \in [k]} \left(\Delta \cdot w(V_0^{(i)}) + \sum_{j \geq 1} \sum_{x \in V_j^{(i)}} w_x d(x_{j-1}^{(i)}, x)\right) \overset{\text{Lemma B.16}}{\leq} 2k^2 \Delta + 42 \sum_{i \in [k]} \sum_{j \geq 1} \mathsf{opt}_1(V_{j-1}^{(i)} \cup V_j^{(i)})$$

$$\leq 2k^2 \Delta + 84 \sum_{i \in [k]} \sum_{x \in V^{(i)}} w_x d(c_i, x)$$

$$= O\left(k^2 \Delta + \sum_{t=1}^{T} \mathsf{Cost}_t(y^*)\right).$$

where we used $\mathsf{opt}_1(V_{j-1}^{(i)} \cup V_j^{(i)}) \leq \sum_{x \in V_{j-1}^{(i)} \cup V_j^{(i)}} w_x d(c_i, x)$. This proves the lemma. $\qquad \square$

Combining Lemma 3.2 and Lemma 3.3, since there are at most $O(k \log T)$ phases, we get:

**Theorem B.17.** *Let $y^*$ be an arbitrary integral vector in $K_T$. Then,*

$$\sum_{t=1}^{T} \mathsf{Cost}_t(y_t) = O\left(\sum_{t=1}^{T} \mathsf{Cost}_t(y^*)\right) + O\left(k^3 \Delta \cdot \sqrt{T} \log(T) \log(kT)\right).$$

## C   Details about the Rounding Algorithms

In this section, we give a detailed description of the online rounding algorithms described in Section 4 for the LEARN-BOUNDED-MEDIAN problem. Let $y_t$ be the fractional solution computed by Algorithm 1 before the arrival of the $t^{th}$ instance $R_t$. Our rounding algorithms will take the fractional solution $y_t$ as input and return an integer solution $Y_t$ with exactly $k$ centers. The *rounding loss* is defined as the ratio of the (expected) cost of the integer solution $Y_t$ to that of the fractional solution $y_t$.

We give two rounding algorithms. The first algorithm is deterministic and has a rounding loss of $O(k)$. The second algorithm is randomized and has only $O(1)$ rounding loss. These are adaptations of existing rounding algorithms for $k$-median [Charikar et al., 1999, Charikar and Li, 2012]; the main difference is that we need to bound the rounding loss for points we haven't seen as well.

### C.1   Deterministic Rounding Algorithm

Initially, there are no centers in $Y_t$. The algorithm considers each point in $R_{<t} = R_0 \cup \ldots R_{t-1}$ in non-decreasing order of their connection costs $D(y_t, i)$, and decides whether to open a center at the current point. A new center is opened at the current point (call it $i$) if the connection cost of $i$ to the previously opened centers in $Y_t$ exceeds $(2k + 2)$ times its connection cost in the fractional solution $y_t$. I.e., $Y_t \leftarrow Y_t \cup \{i\}$ if $D(Y_t, i) > (2k + 2) \cdot D(y_t, i)$.

We show in the next lemma that the algorithm produces a feasible solution, i.e., at most $k$ centers are opened in $Y_t$.

**Lemma C.1.** *The deterministic algorithm produces a feasible solution, i.e., opens at most $k$ centers.*

*Proof.* For $j \in Y_t$, let $B_j = \{x \in R_{<t} : d(j, x) \le (k+2) \cdot D(y_t, j)\}$ be the ball around $j$ of radius $k + 2$ times the fractional cost of $j$. We will show that

    (a) The fractional mass on points in $B_j$ in $y_t$ is strictly greater than $1 - \frac{1}{k+1}$.

    (b) The balls $B_j$ are disjoint for different $j$.

Given these properties, if $|Y_t| \ge k + 1$, then the total fractional mass $y_t$ in the balls $B_j$ for $j \in Y_t$ is strictly greater than $k$, which is a contradiction.

We show the two properties. For property (a), if the fractional mass $y_t$ in $B_j$ is at most $1 - \frac{1}{k+1}$, then $j$ must be served at ar least $\frac{1}{k+1}$ fraction by centers that are outside $B_j$. It follows that $D(y_t, j) \ge \frac{1}{k+1} \cdot (k+2) \cdot D(y_t, j)$, which is a contradiction. For property (b), suppose $B_j, B_{j'}$ overlap where $j' > j$, i.e., $D(y_t, j') \ge D(y_t, j)$. Then, $d(j, j') \le 2(k+2) \cdot D(y_t, j')$, which means that the algorithm will not open a center at $j'$, a contradiction. $\qquad\square$

For the rounding loss, note that the algorithm explicitly ensures that for any $i \in R_{<t}$, we have $D(Y_t, i) \le (2k+2) \cdot D(y_t, i)$. The next claim bounds the rounding loss for points $j \in R_t$.

**Lemma C.2.** *For any point $j \in R_t$, we have $D(Y_t, j) \le (4k + 3) \cdot D(y_t, j)$.*

*Proof.* Let $i$ be the point in $R_{<t}$ that is closest to $j$. Clearly,

$$D(y_t, j) \ge d(i, j),$$

since the entire fractional mass of $y_t$ is on points in $R_{<t}$. Moreover,

$$D(y_t, i) \le D(y_t, j) + d(i, j),$$

since the RHS is at most the cost of serving $i$ using $j$'s fractional centers. Adding the two inequalities, we get $D(y_t, j) \ge \frac{D(y_t, i)}{2}$. Note that there is a center in $Y_t$ that is within distance $(2k+2) \cdot D(y_t, i)$ of $i$ since $i \in R_1 \cup R_2 \cup \ldots \cup R_{t-1}$. The distance from this center to $j$ is at most

$$(2k+2) \cdot D(y_t, i) + d(i, j) \le (4k+2) \cdot D(y_t, j) + D(y_t, j) = (4k+3) \cdot D(y_t, j). \qquad\square$$

This implies that the rounding loss of the deterministic rounding algorithm is $4k + 3$.

## C.2 A Randomized Rounding Algorithm

The rounding algorithm has two phases. The first phase selects a set of centers $\bar{Y}_t$ that might be larger than $k$. This phase is deterministic. The second phase subselects at most $k$ centers from $\bar{Y}_t$ to form the final solution $Y_t$. This phase is randomized. The guarantee that at most $k$ centers are selected in $Y_t$ holds deterministically, but the cost of the solution will be bounded in expectation. We desribe these two phases below.

**Phase 1:** This phase is similar to the deterministic algorithm in the previous section but with different parameters. Initially, there are no centers in $\bar{Y}_t$. The algorithm considers each point in $R_{<t}$ in non-decreasing order of their connection costs $D(y_t, i)$, and decides whether to open a center at the current point. A new center is opened at the current point (call it $i$) if the connection cost of $i$ to the previously opened centers in $\bar{Y}_t$ exceeds 4 times its connection cost in the fractional solution $y_t$. I.e., $\bar{Y}_t \leftarrow \bar{Y}_t \cup \{i\}$ if $D(\bar{Y}_t, j) > 4 \cdot D(y_t, i)$.

**Phase 2:** For any center $i \in \bar{Y}_t$, let $r_i$ be the distance to its closest center in $\bar{Y}_t$. We define the weight of $i$, denoted $w_i$, as the the total fractional mass in $y_t$ in the open ball of radius $r_i/2$ centered at $i$. I.e.,

$$w_i = \sum_{j : d(i,j) < r_i/2} y_t(j).$$

We start by creating a matching on the centers in $\bar{Y}_t$. The first matched pair is the closest pair of centers in $\bar{Y}_t$. The matching continues iteratively by pairing the two closest unmatched centers in $\bar{Y}_t$

in each step. Eventually, either all centers in $\bar{Y}_t$ are in matched pairs or there is a solitary center that goes unmatched. For any center $i \in \bar{Y}_t$ that is in a matched pair, we denote the center it got matched to by $i'$.

Next, we order the matched pairs arbitrarily to create an induced ordering on the points that satisfies the property that $i, i'$ are adjacent. If there is an unmatched point, it is added at the end of this order. Let us call the ordering $i_1, i_2, \ldots, i_{|\bar{Y}_t|}$. We now associate an interval with each point as follows: for point $i_s$, its associated interval is $I_s := [\sum_{j=1}^{s-1} w_{i_j}, \sum_{j=1}^{s} w_{i_j})$, i.e., the points occupy adjacent intervals on the real line in the given order. Next, we generate a value $\theta$ uniformly at random from $[0, 1)$, and add to $Y_t$ every point $i_s \in \bar{Y}_t$ such that $a + \theta \in I_s$ for some non-negative integer $a$.

First, we establish that the algorithm is correct, i.e., $|Y_t| \leq k$.

**Lemma C.3.** *The output $Y_t$ of the randomized rounding algorithm has at most $k$ centers.*

*Proof.* It is sufficient to show that the sum of weights of the centers in $\bar{Y}_t$ is at most $k$. In turn, this follows if we show that the open balls $B_i := \{j : d(i, j) < r_i/2\}$ are disjoint for different $i \in \bar{Y}_t$. Suppose not; let $d(i_1, i_2) < \frac{r_{i_1} + r_{i_2}}{2}$ for $i_1, i_2 \in \bar{Y}_t$. Then, $r_{i_1} \leq d(i_1, i_2)$ and $r_{i_2} \leq d(i_1, i_2)$ by definition of $r_i$. This is a contradiction. $\square$

Next, we bound the rounding loss of the algorithm. Note that for any point $j \in R_1 \cup R_2 \cup \ldots \cup R_{t-1}$, we have the following explicit property from the construction of $\bar{Y}_t$:

$$D(\bar{Y}_t, j) < 4 \cdot D(y_t, j). \tag{13}$$

Another important property sets a lower bound on the weight of any point:

**Claim C.4.** *For any point $i \in \bar{Y}_t$, we have*

$$w_i \geq 1 - \frac{2 \cdot D(y_t, i)}{r_i} > \frac{1}{2}.$$

*Therefore, for any matched pair $i, i' \in \bar{Y}_t$, at least one of $i, i'$ is in $Y_t$.*

*Proof.* For the first inequality, note that $D(y_t, i) \geq 0 \cdot w_i + (r_i/2) \cdot (1 - w_i)$ since at least $1 - w_i$ fraction of the service for $i$ in the fractional solution $y_t$ comes from outside the open ball of radius $r_i/2$ centered at $i$. Re-arranging this inequality gives the first inequality.

For the second inequality, let $j$ be the closest center to $i$ in $\bar{Y}_t$ at the time that $i$ was considered in phase 1 of the rounding algorithm. Since $i$ was added to $\bar{Y}_t$, it must be that $d(i, j) > 4 \cdot D(y_t, i)$. Moreover, any center $\ell$ added to $\bar{Y}_t$ later in the algorithm must satisfy $d(i, \ell) > 4 \cdot D(y_t, \ell) \geq 4 \cdot D(y_t, i)$. Therefore, $r_i > 4 \cdot D(y_t, i)$, which implies the second inequality.

Therefore, for any matched pair $i, i'$, their cumulative weight $w_i + w_{i'} > 1$. This implies that at least one of $i, i'$ will be added to $Y_t$ in phase 2. $\square$

We use these property to bound the rounding loss for $Y_t$:

**Lemma C.5.** *The following properties hold:*

1. *For any point $i \in \bar{Y}_t$, we have $\mathbb{E}[D(Y_t, i)] \leq 4 \cdot D(y_t, i)$.*

2. *For any point $j \in R_0 \cup \ldots \cup R_{t-1}$, we have $\mathbb{E}[D(Y_t, j)] \leq 8 \cdot D(y_t, j)$.*

3. *For any $j \in R_t$, we have $\mathbb{E}[D(Y_t, j)] \leq 17 \cdot D(y_t, j)$.*

*Proof.* We prove each of these properties separately. Note that $\Pr[i \in Y_t] \geq \min(w_i, 1)$ in phase 2 of the algorithm. Using Claim C.4, we have

$$\Pr[i \in Y_t] \geq 1 - \frac{2 \cdot D(y_t, i)}{r_i}.$$

Moreover, with probability 1, we have $D(Y_t, i) \leq 2r_i$. This follows from two cases: (a) If $i$ is matched to its closest point in $\bar{Y}_t$, then by Claim C.4, at least one of $i, i'$ is in $Y_t$. (b) Otherwise, suppose $i$ is unmatched or it is matched to $i'$ which is not its closest point in $\bar{Y}_t$. Let $j$ be closest point

to $i$ in $\bar{Y}_t$. Then, the fact that $j$ was matched to $j'$ and not to $i$ implies that $d(j, j') \le d(i, j) = r_i$. By triangle inequality, $d(i, j') \le 2r_i$, and at least one of $j, j'$ is in $Y_t$ by Claim C.4.

Combining the two observations above, we get

$$\mathbb{E}[D(Y_t, i)] \le \frac{2 \cdot D(y_t, i)}{r_i} \cdot 2r_i = 4 \cdot D(y_t, i).$$

Next, we show the second property. Consider a point $j \in R_{<t} \setminus \bar{Y}_t$. since $j$ was not added to $\bar{Y}_t$, there exists $i \in \bar{Y}_t$ such that $D(\bar{Y}_t, i) \le D(\bar{Y}_t, j)$ and $d(i, j) \le 4 \cdot D(y_t, j)$. Therefore,

$$\mathbb{E}[D(Y_t, j)] \le d(i, j) + \mathbb{E}[D(Y_t, i)] \le 4 \cdot D(y_t, j) + 4 \cdot D(y_t, i) \le 8 \cdot D(y_t, j),$$

where we used the first property in the second to last inequality.

Finally, we show the third property. Consider a point $j \in R_t$. Let $i$ be the point in $R_{<t}$ that is closest to $j$. Clearly,

$$D(y_t, j) \ge d(i, j),$$

since the entire fractional mass of $y_t$ is on points in $R_{<t}$. Moreover,

$$D(y_t, i) \le D(y_t, j) + d(i, j),$$

since the RHS is at most the cost of serving $i$ using $j$'s fractional centers. Adding the two inequalities, we get $D(y_t, i) \le 2 \cdot D(y_t, j)$. Therefore,

$$\mathbb{E}[D(Y_t, j)] \le \mathbb{E}[D(Y_t, i)] + d(i, j) \le 8 \cdot D(y_t, i) + D(y_t, j) \le 17 \cdot D(y_t, j),$$

where we used the second property in the second to last inequality. $\square$

This implies that the rounding loss of the randomized rounding algorithm is 17.

## D   Final Bounds

We now put together our reduction, fractional algorithm, and rounding procedure to obtain deterministic and randomized algorithms for LEARN-MEDIAN. By combining Theorem 2.1, Theorem B.17 and Lemma C.2, we have the following theorem.

**Theorem D.1.** *We give a deterministic algorithm for* LEARN-MEDIAN *with the following performance bound:*

$$\sum_{t=1}^{T} \rho(Y_t, V_t) \le O(k) \cdot \sum_{t=1}^{T} \rho(Y^*, V_t) + O\left(k^4 \Delta \cdot \sqrt{T} \log(T) \log(Tk)\right)$$

*where $Y^*$ is the best fixed solution in hindsight.*

Similarly, by combining Theorem 2.1, Theorem B.17 and Lemma C.5, we have the following theorem.

**Theorem D.2.** *We give a randomized algorithm for* LEARN-MEDIAN *with the following expected performance bound:*

$$\sum_{t=1}^{T} \mathbb{E}[\rho(Y_t, V_t)] \le O(1) \cdot \sum_{t=1}^{T} \rho(Y^*, V_t) + O\left(k^3 \Delta \cdot \sqrt{T} \log(T) \log(Tk)\right)$$

*where $Y^*$ is the best fixed solution in hindsight.*

## E   Detailed Experiments

In this section, we give a detailed description of the experiments in Section 5. We use a heuristic to improve the performance of the rounding algorithms in all our experiments, where we do a binary search over the best threshold on the distance to the already opened centers. For the deterministic rounding algorithm for LEARN-BOUNDED-MEDIAN, we open a new center at $i$ if its distance to the already opened centers exceeds $(2k + 2)D(y_t, i)$. We can always slacken this factor of $(2k + 2)$ to

something lower as long as the total number of centers opened does not exceed $k$, without worsening the theoretical approximation ratio. Similarly in the randomized rounding algorithm for LEARN-BOUNDED-MEDIAN, we can slacken the threshold of $4D(y_t, i)$ in the initial filtration step, as long as the fractional mass in the balls we later consider are at least $1/2$.

We compute optimal-in-hindsight solutions for the instances using Gurobi's ILP solver, accessed under an academic license. Unless mentioned otherwise, experiments were conducted using Google Colab (code available at `https://github.com/neurips2025-colab/neurips2025`), using four Intel® Xeon® CPUs (2.20 GHz) with 13 GB of RAM each, running in parallel over about 24 compute hours. In all our experiments, the underlying metric is the Euclidean metric. Unless mentioned otherwise, we plot the ratio between $\sum_{\tau=1}^{t} \rho(Y_\tau, V_\tau)$ and $\sum_{\tau=1}^{t} \rho(Y^*, V_\tau)$ in our approximation ratio plots, where $Y^*$ is the optimal-in-hindsight solution for the entire input sequence.

**Uniform Square:** In this example, the underlying metric space consists of $400$ uniformly random points in the unit square $[0, 1] \times [0, 1]$. 10 points arrive each round, chosen uniformly at random, and we run the experiment with a time horizon of $T = 1000$. We compare the cost of the optimal solution with the deterministic and randomized algorithms, as well as the intermediate fractional solution that we maintain. We generate 10 random instances in total, and report the standard deviation and mean of the approximation ratio over time. For the randomized algorithm, we first average the approximation ratio for each instance over 5 random runs of the rounding algorithm. We run this experiment with $k = 2, 3, 6$.

As we see in Fig. 3, both the randomized and deterministic algorithms converge to natural solutions, distributed roughly uniformly over the unit square. The approximation ratio also approaches 1, especially after the influence of the initial additive regret declines.

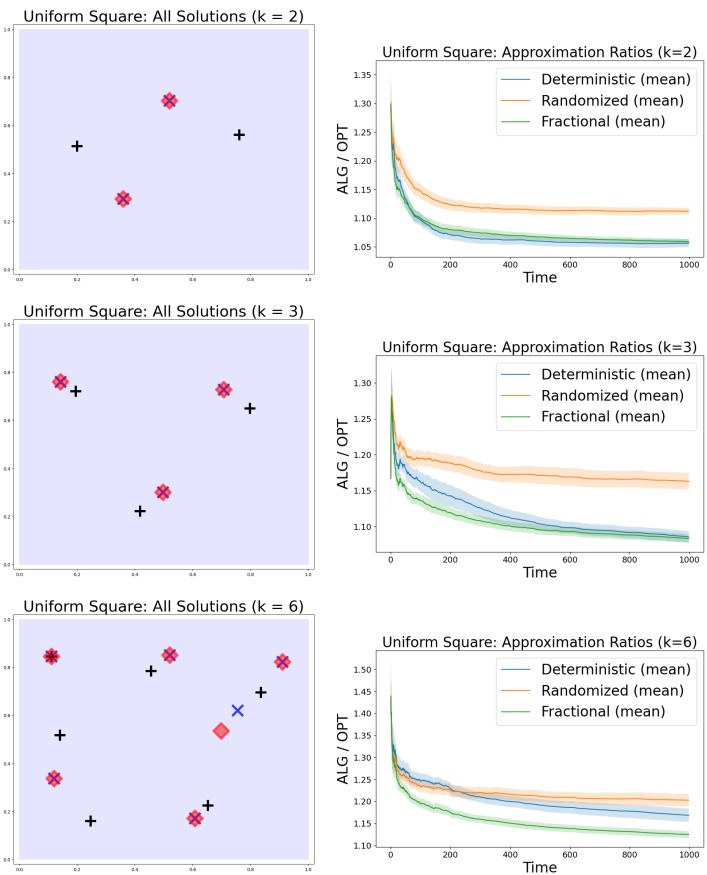

Figure 3: **(Uniform Square):** The optimal (black plus), deterministic (blue cross), and randomized (red diamond) solutions for one of the random instances (left) and approximation ratios – avg. and std. dev. over 10 random instances (right) for $k = 2, 3, 6$.

**Uniform Rectangle:** In this example, the underlying metric space consists of $400$ uniformly random points in the rectangle $[0, 1] \times [0, 10]$. $10$ points arrive each round, chosen uniformly at random, and we run the experiment with a time horizon of $T = 1000$. We compare the cost of the optimal solution with the deterministic and randomized algorithms, as well as the intermediate fractional solution that we maintain. We generate $10$ random instances in total, and report the standard deviation and mean of the approximation ratio over time. For the randomized algorithm, we first average the approximation ratio for each instance over $5$ random runs of the rounding algorithm. We run this experiment with $k = 2, 3, 6$.

As we see in Fig. 4, both the randomized and deterministic algorithms converge to natural solutions, distributed roughly uniformly over the rectangle. The approximation ratio also approaches $1$, especially after the influence of the initial additive regret declines.

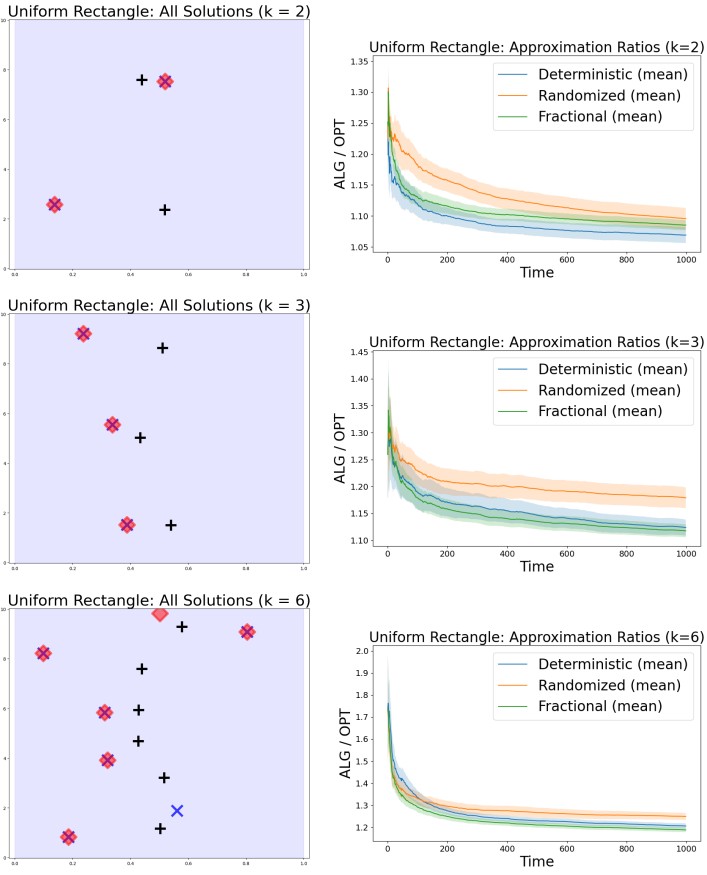

Figure 4: **(Uniform Rectangle):** The optimal (black plus), deterministic (blue cross), and randomized (red diamond) solutions for one of the random instances (left) and approximation ratios – avg. and std. dev. over 10 random instances (right) for $k = 2, 3, 6$.

**Multiple Clusters:** In this example, the underlying metric space consists of $k$ clusters with center in the unit square $[0, 1] \times [0, 1]$. The cluster centers are chosen uniformly at random from the unit square, and we then generate points in a radius of $0.05$ around each cluster center, for a total of about $400$ points. $20$ points arrive each round, chosen uniformly at random, and we run the experiment with a time horizon of $T = 1000$. We compare the cost of the optimal solution with the deterministic and randomized algorithms, as well as the intermediate fractional solution that we maintain. We generate $10$ random instances in total, and report the standard deviation and mean of the approximation ratio over time. For the randomized algorithm, we first average the approximation ratio for each instance over $5$ random runs of the rounding algorithm. We run this experiment with $k = 4, 8, 12, 16$.

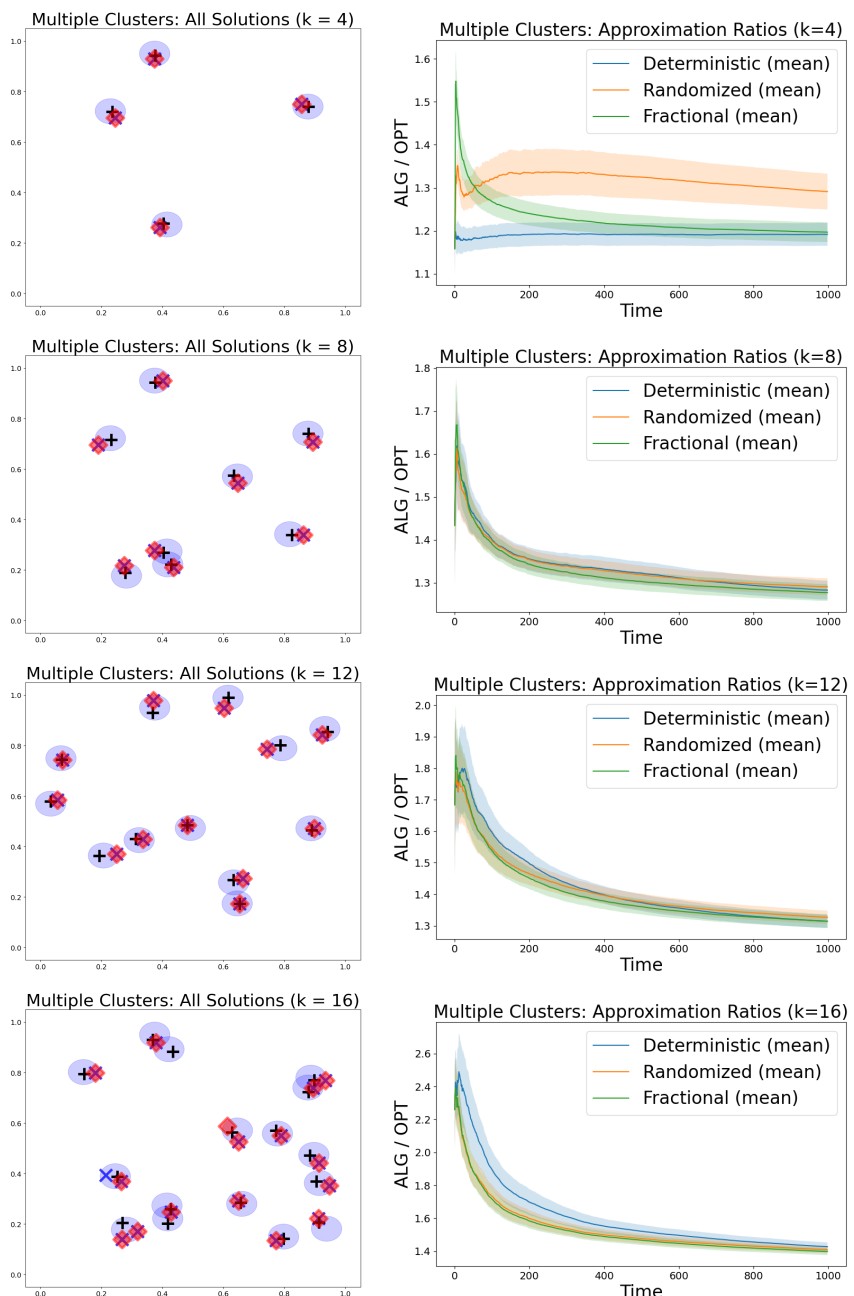

Figure 5: (**Multiple Clusters**): the optimal (black plus), deterministic (blue cross), and randomized (red diamond) solutions for one of the random instances (left) and approximation ratios - avg. and std. dev. over 10 random instances (right) for $k = 4, 8, 12, 16$.

Once again, as we see in Fig. 5, both the randomized and deterministic algorithms converge to natural solutions, with centers distributed over the different clusters. The approximation ratio also approaches 1, especially after the influence of the initial additive regret declines.

**Uniform Hypersphere:** In this example, $k = 1$ and the underlying metric space consists of 400 points: one point is the origin and the rest are chosen uniformly at random from the unit hypersphere. We run the experiment with a time horizon of $T = 2000$. For the first 100 rounds, each instance consists of 10 points chosen at random from the boundary of the hypersphere. However, in the next round we also include the origin in the instance, thus revealing the origin to our algorithm. We then

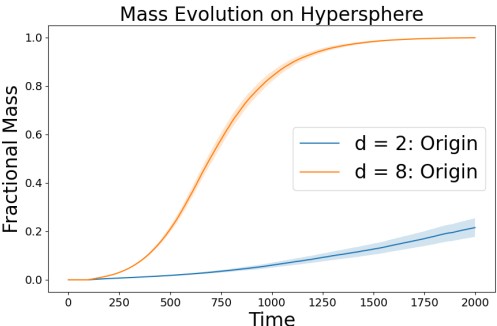

Figure 6: **(Uniform Hypersphere):** Fractional mass of algorithm over time at the center (avg. and std. dev. over 10 runs), $d = 2, 8$

continue generating instances as before. We do this for for 2 different dimensions of the hypersphere ($d = 2, 8$). We generate 10 random instances in total, and report the standard deviation and mean of the total mass that the fractional algorithm places on the origin.

As we see in Fig. 6, the fractional mass is initially fully on the boundary, but once we introduce the origin the mass slowly shifts to being just on the origin. This happens because the origin is a better solution for the instance sequence compared to any solution solely on the boundary. Moreover, this transition is more rapid for higher dimensions, as typical distance between 2 random points on the hypersphere increases with dimension. On the other hand, the distance from the origin to the hypersphere boundary is always 1.

**Oscillating Instances:** In this example, the underlying metric space consists of 2 clusters with 10 points each. The 2 clusters are generated by picking points uniformly at random from squares of side length $0.4$ centered at $(0, 0)$ and $(1, 0)$ respectively, and we run the experiment with a time horizon of $T = 3^5$. The instance sequence alternates between the first and second cluster, where we give the first cluster for the first 3 rounds, then the second cluster up to the $9^{th}$ round, then the first cluster up to the $27^{th}$ round, and so on. One difference in this example is that we compare our performance against the dynamically optimal fractional solution as opposed to a fixed optimal-in-hindsight integer solution: that is, for each time step we compare against the best fixed solution for the instances so far. That is, we plot the ratio between $\sum_{\tau=1}^{t} \rho(y_\tau, V_\tau)$ and $\sum_{\tau=1}^{t} \rho(\mathsf{OPT}_t, V_\tau)$, where $\mathsf{OPT}_t$ is the best fixed fractional solution for instances $V_1, ..., V_t$ and $y_\tau$ is the intermediate fractional solution we maintain. We also study how the fractional mass of the (dynamic) optimal solution and the algorithm in the first cluster changes over time in order to obtain a qualitative understanding of how the algorithm shifts its centers towards new points that arrive. We run this experiment with $k = 1, 2, 3, 4$.

As we see in Fig. 7, $k = 1$, the algorithm starts shifting fractional mass before the optimal solution. Consequently, our algorithm ends up outperforming the optimal solution in the time horizon that we consider, as seen from the approximation ratio plots. For $k = 2, 3, 4$, we observe that the algorithm quickly moves a unit of mass to the cluster that is currently arriving, and then slows down as the gain in increasing the fractional mass on the cluster is minimal.

**Scale Changes:** In this example, the underlying metric space consists of 5 clusters with 10 points each. The clusters are generated by picking points uniformly at random from squares of side length $0.4$ centered at $(\lfloor 10^{i-1} \rfloor, 0)$ for $i = 0, 1, 2, 3, 4$, and we run the experiment with a time horizon of $T = 3^5$. In this example, we have 5 clusters with 10 points each, with a time horizon of $T = 3^5$. The instance sequence iterates through the clusters, where we give the first cluster for the first 3 rounds, then the second cluster up to the $9^{th}$ round, and so on. We study how the fractional mass of the algorithm in the cluster that is currently arriving changes over time in order to obtain a qualitative understanding of how the algorithm shifts its centers toward new far away points that arrive. The approximation ratios plotted are between $\sum_{\tau=1}^{t} \rho(y_\tau, V_\tau)$ and $\sum_{\tau=1}^{t} \rho(\mathsf{OPT}_t, V_\tau)$, where $\mathsf{OPT}_t$ is the best fixed fractional solution for instances $V_1, ..., V_t$ and $y_\tau$ is the intermediate fractional solution we maintain. We run this experiment with $k = 1, 2, 3$.

As we see in Fig. 8, we once again see that for $k = 1$, the fractional algorithm ends up outperforming the optimal solution. For $k = 2, 3$, we observe that the algorithm quickly moves a unit of mass to the

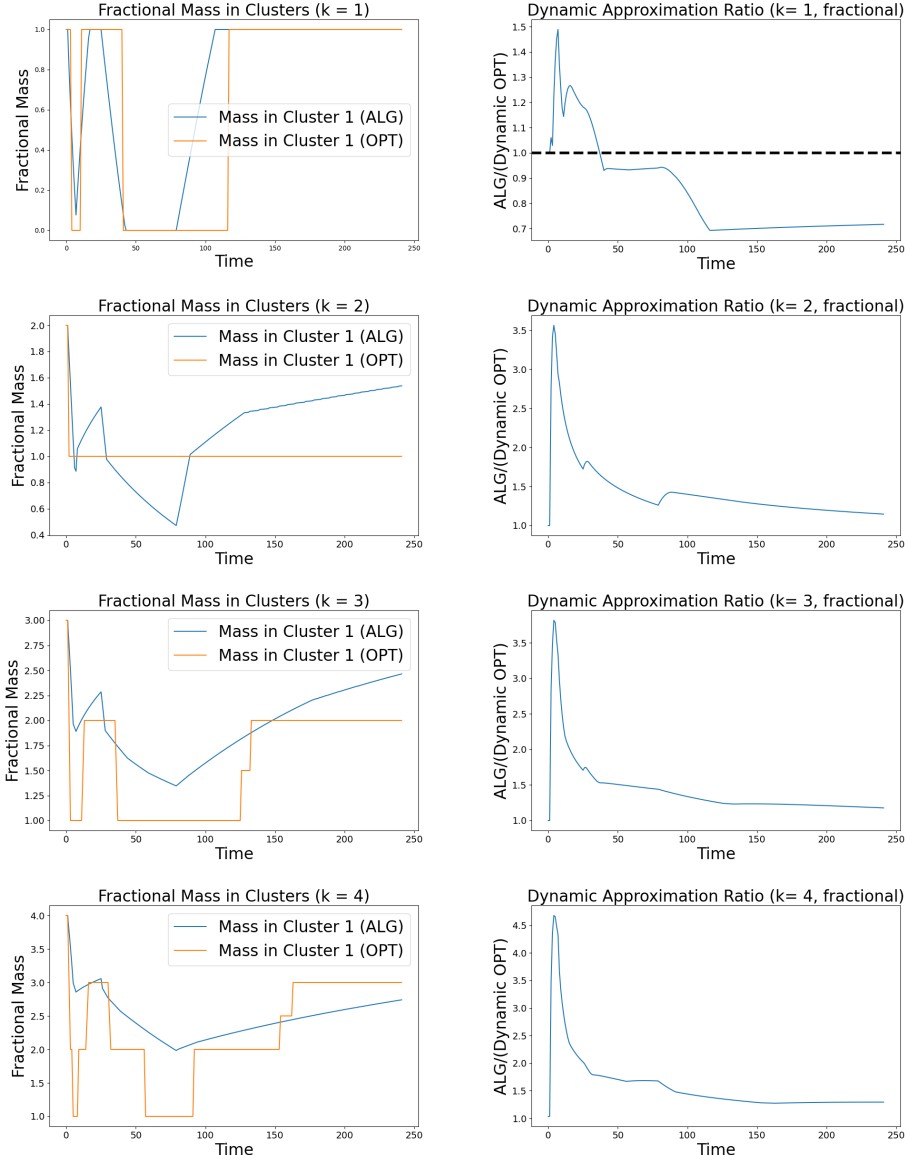

Figure 7: **(Oscillating Instances):** Fractional mass of algorithm and OPT over time in one of the clusters (left) and corresponding dynamic approximation ratios (right), for $k = 1, 2, 3, 4$.

cluster that is currently arriving, and then slows down rapidly as the gain in increasing the fractional mass on the cluster is very minimal. The change is even more rapid in this case compared to the previous experiment due to the large scale changes which lead to a larger gradient norm, and hence a more conservative learning rate.

One question that arises from the approximation ratio plots for $k = 2, 3$ is the large approximation factor, but this can be explained by the additive term that is quite large in this case. For $k > 1$, the dynamic optimal solution immediately shifts a center from the previous clusters to the new one as it is a much better solution for the sequence so far, while the algorithm accrues a large cost until it shifts a mass of one into the new cluster. To illustrate this, we run the experiment again for $k = 2, 3$ with 4 clusters and a time horizon of $T = 50000$ (we only use 4 clusters this time due to computational reasons, the time required for the additive term to be insignificant grows with $\Delta$), where we continue giving the $4^{th}$ cluster even after the first 81 rounds. We see that the approximation ratio eventually approaches 1, see Fig. 9.

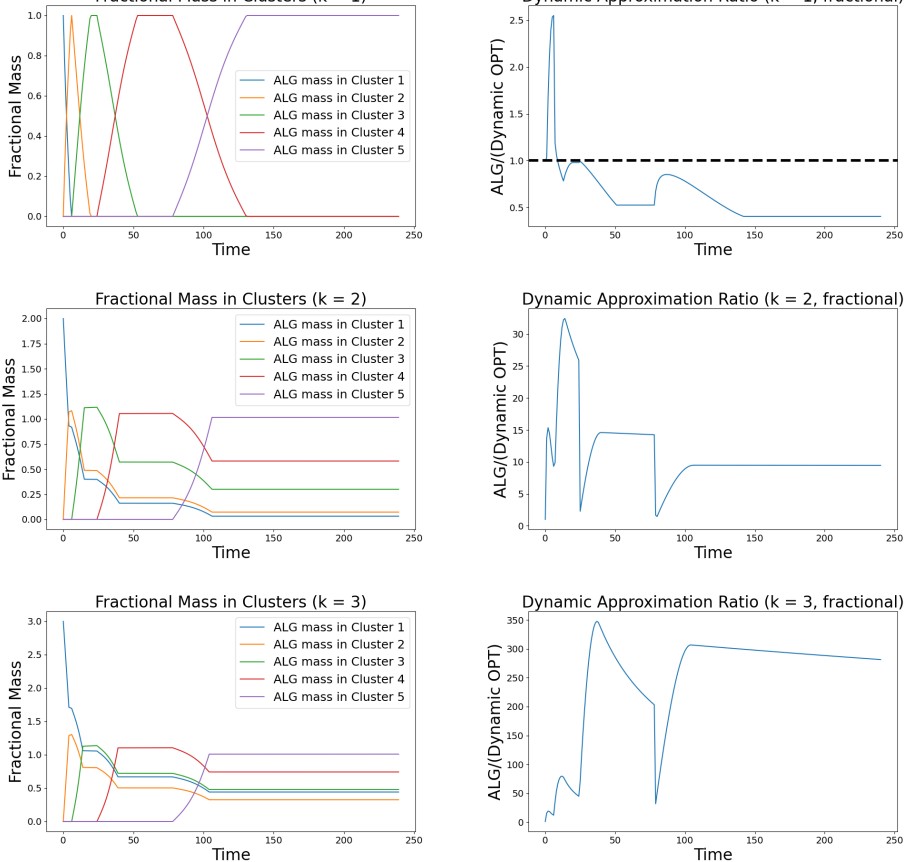

Figure 8: **(Scale Changes):** Fractional mass of algorithm over time in one of the clusters (left) and corresponding dynamic approximation ratios (right), for $k = 1, 2, 3$.

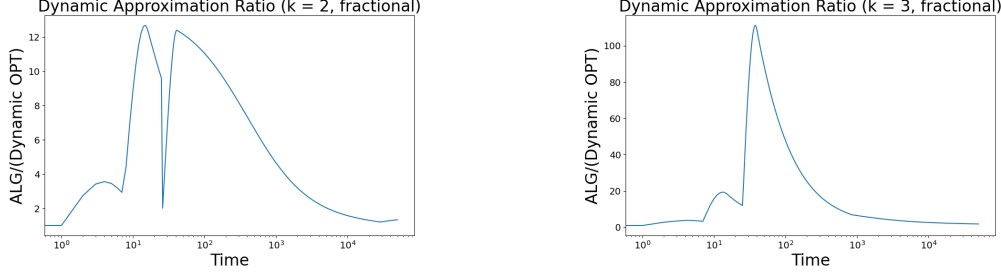

Figure 9: **(Scale Changes):** Dynamic approximation ratio plots with $T = 50000$, for $k = 2, 3$.

**Small Drifts:** In this example, the metric space consists of 2500 points, and we run the experiment with a time horizon of $T = 250$. The underlying instances are created by randomly drawing 10 points (out of which 5 random points arrive in a round) in a disc centered around the origin, but then shifting the center of the disc to the right by $0.02$ for each instance. We compare the cost of the optimal solution with the deterministic and randomized algorithms, as well as the intermediate fractional solution that we maintain. We generate 10 random instances in total, and report the standard deviation and mean of the approximation ratio over time. For the randomized algorithm, we first average the approximation ratio for each instance over 5 random runs of the rounding algorithm. We run this experiment with $k = 1, 2, 3$.

As we see in Fig. 10, the final average approximation ratios were less than 2 in all cases. (In contrast, if the algorithm were to use the initial solutions throughout, i.e., does not automatically adapt to the

drift, then the cost ratios in each round are around 5-15.) The approximation ratio is small initially as we compare against the optimal-in-hindsight solution for the entire input sequence.

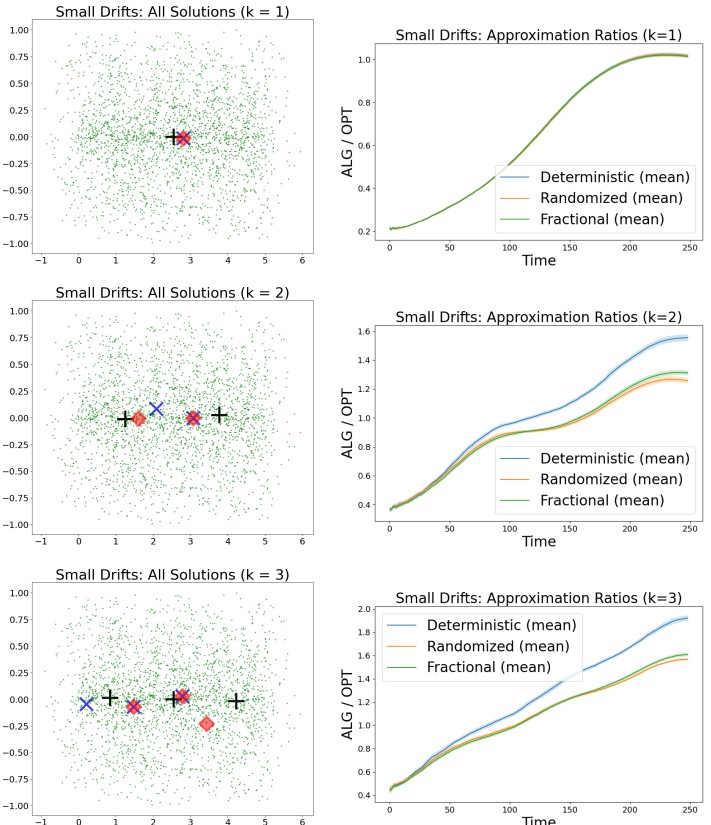

Figure 10: **(Small Drifts):** The optimal (black plus), deterministic (blue cross), and randomized (red diamond) solutions for one of the random instances (left) and approximation ratios – avg. and std. dev. over 10 random instances (right) for $k = 1, 2, 3$.

Finally, note that the time complexity of our learning algorithm (besides the reduction step) is $O(k^2 T^3)$, independent of the number of points $n$. To empirically confirm scalability, we ran experiments with $n = 1000, 2000, \dots, 6000$ while keeping $k, T$ fixed ($k = 5, T = 100$) and observed that the running time of the learning algorithm (besides the reduction step) was always 2-3 seconds on Google Colab (2.20 GHz Intel® Xeon® CPU, 51 GB RAM).

# F Lower Bounds

In Theorem D.1 we gave an efficient deterministic algorithm for LEARN-MEDIAN with a multiplicative $O(k)$ approximation factor and sublinear regret. We now give an information theoretic lower bound that states that *any* algorithm for LEARN-MEDIAN with sublinear regret must incur an $\Omega(k)$ multiplicative factor loss.

**Theorem F.1.** *Any deterministic algorithm for the* LEARN-MEDIAN *problem cannot achieve a multiplicative $o(k)$ approximation guarantee with an additive regret term that is sublinear in $T$. That is, any bound as below is impossible*

$$\sum_{t=1}^{T} \mathbb{E}[\rho(Y_t, V_t)] \leq o(k) \cdot \sum_{t=1}^{T} \rho(Y, V_t) + o(T)f(k, n, \Delta) \tag{14}$$

*where $Y$ is any fixed solution for* LEARN-MEDIAN. *Moreover, this holds even when the metric space is known upfront.*

*Proof.* The lower bound result is inspired by the competitive ratio lower bound for deterministic online paging. Suppose for the sake of contradiction that there is an online algorithm $\mathcal{A}$ that outputs a solution $Y_t$ at time $t$ and satisfies the condition (14). We now describe our hard instance $\mathcal{M} = (V, d)$. The underlying metric space $\mathcal{M}$ consists of $n = 2(k+1)$ points. It contains $k+1$ different clusters of 2 points each, such that the distance between any 2 points belonging to different clusters is $\Delta \gg 1$. The distance between any 2 points in the same cluster is 1.

We construct our sequence of sub-instances $V_0, \ldots, V_T$ in an adversarial fashion: consider a round $t$, and let $Y_t$ be the set of $k$ centers outputted by $\mathcal{A}$ in round $t$. Since there are $k+1$ clusters in the metric space, it follows that there is always at least 1 cluster which does not overlap with $Y_t$. Thus, given $Y_t$, define $V_t$ to consist of this cluster and any other $k-1$ clusters. For the round zero, we just take $V_0$ to be the entire metric space.

Clearly, $\mathcal{A}$ must suffer a cost of at least $\frac{2\Delta}{k}$ for each round after round zero. This is because the optimal solution for $V_t$ places exactly 1 point in each of the $k$ clusters with a total connection cost of $k$, while $Y_t$ must have a connection cost of at least $2\Delta$ as it doesn't place any center in one of the clusters present in $V_t$.

Let us now examine the optimal fixed solution in hindsight for this adversarial input sequence. The optimal solution places 1 center each in the $k$ clusters that appear the most frequently. Thus, it suffers a total cost at most $\frac{2\Delta T}{(k+1)k}$. Putting it together and using (14), we have

$$\frac{2\Delta T}{k} \leq o(k) \cdot \frac{2\Delta T}{(k+1)k} + o(T)f(k, n, \Delta)$$

or equivalently,

$$2\Delta(k+1) \leq o(k) \cdot 2\Delta + \frac{o(T)}{T}k(k+1)f(k, n, \Delta)$$

which gives a contradiction as $T \to \infty$ as $k, n, \Delta$ are fixed parameters of the underlying metric space. $\square$

We now extend the above result to the setting where the online algorithm is randomized: again, we show that the result in Theorem D.2 is essentially tight, i.e., any randomized algorithm with sublinear additive regret must incur a constant factor multiplicative loss.

**Theorem F.2.** *Any (randomized) algorithm for the* LEARN-MEDIAN *problem cannot achieve a multiplicative $(1 + \varepsilon)$ approximation guarantee with an additive regret term that is sublinear in $T$. That is, any bound as below is impossible*

$$\sum_{t=1}^{T} \mathbb{E}[\rho(Y_t, V_t)] \leq (1+\varepsilon) \cdot \sum_{t=1}^{T} \rho(Y, V_t) + o(T)f(k, n, \Delta) \tag{15}$$

*where $0 \leq \varepsilon < 1$ is an arbitrary constant and $Y$ is any fixed solution for* LEARN-MEDIAN.

*Proof.* Suppose for the sake of contradiction that there is an algorithm $\mathcal{A}$ that satisfies (15) for some constant $0 \leq \varepsilon < 1$. We first describe our hard instance $\mathcal{M} = (V, d)$. Our metric space $\mathcal{M}$ consists of $n = mk$ points, $m \gg k$. It contains $k$ different clusters of $m$ points each, such that the distance between any 2 points belonging to 2 different clusters is $\Delta \gg k$. The clusters themselves are star graphs with $m - 1$ leaves and a center point, the distance between any 2 leaves being 2 and the center-leaf distance being 1.

We construct our sequence of sub-instances $V_0, \ldots, V_T$ in an adversarial fashion. In round zero, we just take $V_0$ to be all the leaves. For $t = 1, \ldots, T - 1$, $V_t$ consists of 2 randomly chosen leaves from each cluster. For the last round, we take $V_T$ to be all the cluster centers and one other leaf.

We claim that the algorithm $\mathcal{A}$ must suffer an expected loss of at least $4(1 - 3/m)$ in each round from 1 to $T - 1$. Let $Y_t$ be the subset of size $k$ selected by $\mathcal{A}$ in round $t$. If $Y_t$ does not contain a point from each of the clusters it has a connection cost of at least $\Delta$ for round $t$; otherwise $Y_t$ contains exactly one point from each cluster. For a given cluster, the probability that the point in $Y_t$ doesn't belong to the two points of this cluster in $V_t$ is at least $(1 - 1/m)(1 - 2/m) \geq 1 - 3/m$. Thus linearity of expectation implies that the expected connection cost of $Y_t$ is at least $\min\{\Delta, 4k(1 - 3/m)\} \geq 4k(1 - 3/m)$. On the other hand, the optimal connection cost for $V_t$, $1 \leq t \leq T - 1$ is $k$, corresponding to picking 1 of the chosen leaves from each of the clusters.

Let us now examine the optimal fixed solution in hindsight for this adversarial input sequence. We claim that it suffers a total loss of at most $2T$. Indeed, consider the fixed solution $Y$ that picks each of the cluster centers, it suffers a loss of $\frac{1}{1} = 1$ in the last round as it is the optimal solution for $V_T$, and suffers a loss of at most $\frac{2k}{k} = 2$ for each round from 1 to $T - 1$. This is because its connection cost is at most $2k$, and the optimal connection cost for $V_t$, $1 \leq t \leq T - 1$ is $k$, corresponding to picking 1 of the chosen leaves from each of the clusters. Putting it together, we obtain

$$4(T - 1)(1 - 3/m) \leq (1 + \epsilon) \cdot 2T + o(T)f(k, n, \Delta)$$

or equivalently,

$$2 - 6/m \leq (1 + \epsilon) \cdot \frac{T}{T - 1} + \frac{o(T)}{2(T - 1)}f(k, n, \Delta)$$

which gives a contradiction by taking $m$ large enough that $2 - 6/m > 1 + \epsilon$ and then taking $T \to \infty$.

$\square$

Finally, we show that an additive loss of $\Omega(k\Delta)$ is unavoidable if the multiplicative loss is a constant.

**Theorem F.3.** *Any (randomized) algorithm for the* LEARN-MEDIAN *problem must suffer* $\Omega(k\Delta)$ *additive loss even when allowing for* $O(1)$ *multiplicative error. That is, any bound as below is impossible*

$$\sum_{t=1}^{T} \mathbb{E}[\rho(Y_t, V_t)] \leq O(1) \cdot \sum_{t=1}^{T} \rho(Y, V_t) + o(k\Delta) \tag{16}$$

*where $Y$ is any other fixed solution for* LEARN-MEDIAN.

*Proof.* Suppose for the sake of contradiction that there is an algorithm $\mathcal{A}$ satisfying (16). We first describe our hard instance $\mathcal{M} = (V, d)$. The metric space $\mathcal{M}$ consists of $n = mk$ points, $m \gg k$. It contains $k$ different clusters of $m$ points each, such that the distance between any 2 points belonging to different clusters is $\Delta \gg 1$. The distance between any 2 points in the same cluster is 1.

Take $T = k - 1$ and $V_t$ to be the $(t + 1)^{th}$ cluster for $0 \leq t \leq k - 1$. The algorithm $\mathcal{A}$ must suffer a loss of at least $\frac{m\Delta}{m-k} \geq \Delta/2$ for each round $t$ after round zero. This is because it has not seen the $(t + 1)^{th}$ cluster at time $t$ and hence cannot place any center there, thus suffering a connection cost of at least $m\Delta$, and the optimal solution for $V_t$ consists of $k$ points from the same cluster, with a total connection cost of $m - k$. Thus, the total loss suffered by the algorithm is at least $\frac{(k-1)\Delta}{2}$.

On the other hand, the optimal fixed solution in hindsight places 1 center in each of the $k$ clusters, suffering a loss of $\frac{m-1}{m-k} \leq 2$ each round after round zero. Thus, the total loss suffered by the optimal solution is at most $2(k - 1)$. Putting it together, we obtain

$$\frac{(k-1)\Delta}{2} \le O(1) \cdot 2(k-1) + o(k\Delta)$$

which gives a contradiction as $\Delta \to \infty$, as desired. □

**Approximate Follow-the-Leader** So far we have given information theoretic lower bounds on the multiplicative loss and the additive regret term for any online algorithm. Follow The Leader (FTL) is another commonly used algorithm – at each time $t$, output $Y_t$ that minimizes the total cost $\sum_{t'<t} \mathsf{cost}_{t'}(Y_t)$. In our setting, computing the optimal cost (i.e., cost of an optimal $k$-median instance) is NP-hard. Thus, a natural extension of this approach would be Approximate FTL: at each time $t$, output $Y_t$ that is a constant factor approximation to the objective function $\sum_{t'=1}^{t} \rho(Y, V_{t'})$. We show strong lower bounds for this algorithm even when $k = 1$:

**Theorem F.4.** *Approximate FTL cannot achieve a $o(\log \Delta / \log \log \Delta)$ multiplicative approximation guarantee with sublinear regret even when $k = 1$, i.e., any bound as below is impossible:*

$$\sum_{t=1}^{T} \rho(Y_t, V_t) \le o(\log \Delta / \log \log \Delta) \cdot \sum_{t=1}^{T} \rho(Y, V_t) + o(T)f(k, n, \Delta), \tag{17}$$

*where $Y_t$ is the set of centers outputted by Approximate FTL at time $t$ and $Y$ is an arbitrary set of $k$ centers.*

*Proof.* For ease of notation, assume that Approximate FTL uses a 4-approximation algorithm for the (off-line) $k$-median problem – the argument remains similar for any constant approximation algorithm. We shall also fix $k = 1$.

We now describe the metric space $\mathcal{M}$. $\mathcal{M}$ consists of a star graph with $2\lambda$ leaves, where $\lambda$ is a parameter that we shall fix later. At each of the leaves, we have two vertices that are at distance 2 from each other (thus, $\mathcal{M}$ is given by the shortest path metric on a two level tree, where the root has $2\lambda$ children and every node at the first level has two children). Let the children of the root $r$ be labeled $v_1, \ldots, v_{2\lambda}$. The two children of $v_i$ are labeled $v_i^1$ and $v_i^2$. The edge $(r, v_i)$ has length $\Delta_i = \frac{\Delta}{\lambda^i}$, where $\Delta = \lambda^{2\lambda+1}$ (assume $\lambda \gg 1$).

At time $t = 0$, we give the entire metric space as the instance. The subsequent input sequence is divided into $2\lambda$ phases. For each time $t$ in phase $h \in [2\lambda]$, the input $V_t$ consists of the two points $v_h^1$ and $v_h^2$. Phase $h$ lasts for $T_h := \lambda^h T_0$ timesteps, where $T_0 \gg 1$. Note that $T_h \Delta_h = T_0 \Delta$ for all $h \in [2\lambda]$.

The off-line solution $Y$ places a center at the root of the star. Thus, for any time $t$ in phase $h$, $\rho(Y, V_t) = \frac{2(\Delta_h+1)}{2} = \Delta_h + 1$. Thus, we see that

$$\sum_{t=1}^{T} \rho(Y, V_t) = \sum_{h} T_h(\Delta_h + 1) = 2\lambda T_0 \Delta + T \le 3\lambda T_0 \Delta, \tag{18}$$

because $T = \sum_h \lambda^h T_0 \le 2\lambda^{2\lambda} T_0 \le \Delta T_0$ and the number of phases is $2\lambda$.

Now, we estimate the corresponding quantity for Approximate FTL. Consider a phase $h$. The 1-median instance at a time $t$ in this phase consists of $T_{h'}$ points at each of $v_{h'}^1$ and $v_{h'}^2$ for $h' < h$ and a certain number of points at the children of $v_h$. Since $T_h \Delta_h = T_0 \Delta$ for all $h$, we claim that this 1-median instance has optimal cost at least $(h-2)T_0\Delta$. Indeed, consider a 1-median solution that places a center at the root or at $v_j$ or its children. Then all points at $v_{h'}^1$ and $v_{h'}^2$, where $h' \le h-1, h' \ne h$, incur a cost at least $\frac{2T_{h'}\Delta_{h'}}{2} = T_0\Delta$. Now, consider the solution that places a center at $v_{h-1}$ (at any time during phase $h$). Again it is easy to see that its total cost is at most $\frac{4(h-2)T_0\Delta+2(T_{h-1})+4\lambda T_h \Delta_h}{2} \le T_0\Delta(2h-3+2\lambda)$, which is at most $4((h-2)T_0\Delta)$ when $h \ge \lambda+3$, and hence is a 4-approximation for $h \in [\lambda+3, 2\lambda]$

Thus, we can assume that for any phase $h \ge \lambda + 3$, $\mathcal{A}$ outputs $v_{h-1}$ as the 2-approximate 1-median center. Now consider such a phase $h$: at each time $t$ during this phase, optimal cost for $V_t$ is 2, but

the algorithm incurs at least $2\Delta_{h-1} = 2\lambda\Delta_h$. Thus, $\rho(Y_t, V_t) \geq \lambda\Delta_h$. Summing over all times $t$ in this phase, we see that

$$\sum_{t \text{ in phase } h} \rho(Y_t, V_t) \geq \lambda T_h \Delta_h = \lambda T_0 \Delta.$$

Summing over all the phases $h \geq \lambda + 3$ (note that there are $\lambda - 2$ such phases), we get

$$\sum_{t \text{ in phases } h \geq \lambda + 3} \rho(Y_t, V_t) \geq \frac{\lambda^2}{2} T_0 \Delta.$$

The result now follows from the above and inequality (18) (note that $\lambda = \theta(\log \Delta / \log \log \Delta)$ and $T_0$ is a parameter independent of $n, k, \Delta$). $\qquad\square$

