# OpenReview forum: "Learning-Augmented Algorithms for $k$-median via Online Learning"
_NeurIPS.cc/2025/Conference — NeurIPS 2025 poster_

### Official Review · Reviewer_Rx2G · 2025-07-01

**Clarity:** 3
**Significance:** 3
**Originality:** 3
**Rating:** 4
**Confidence:** 4

**Summary:**

This paper introduces LEARN-MEDIAN , a novel framework for learning-augmented algorithms that dynamically adapts to sequences of online instances in metric spaces. Unlike prior work that relies on static predictions, the authors propose an online learning model that iteratively refines a fixed solution to minimize the average approximation ratio over time. The key contributions include:

1.	A new paradigm where the algorithm learns from historical instances to optimize future performance, achieving sublinear regret against the hindsight-optimal solution by using Hyperbolic Entropy Regularizer

2.	A deterministic rounding algorithm with $O(k)$-competitive ratio and $ O(k^4 \Delta\sqrt{T \log T})$ regret, a randomized rounding algorithm with improved $O(1)$-competitive ratio and $ O(k^3 \Delta\sqrt{T \log T})$ regret.

3.	A proof of Lower Bounds for any algorithm for LEARN-MEDIAN with sublinear regret.

4.	Experiments demonstrate the algorithm’s robustness and convergence to near-optimal performance (approximation ratios close to 1).

**Questions:**

1.	The paper focuses on optimizing the average approximation ratio across instances. However, it does not explicitly analyze the algorithm’s Consistency (performance under accurate predictions) or Robustness (worst-case guarantees under prediction errors). Specifically: When the predictions (e.g., gradient estimates from historical instances) are inaccurate or adversarial, can the approximation ratio still be bounded? How does the algorithm’s performance degrade as the quality of predictions deteriorates? A discussion on these aspects would strengthen the theoretical grounding of the work."

2.	The paper's experiments evaluate the algorithm on a unit square metric space ([0, 1] × [0, 1]). However, in real-world applications, the metric space may have irregular scales (e.g., a rectangle [0, 1] × [0, 5] or [0, 1] × [0, 10]). What is the approximation ratio of the algorithm when the metric space is not square? Whether the algorithm is effective on real-world datasets?

3.	In Fig .3 (for k = 2, 3, 6) and 4 (for k = 4), the average approximation ratio of the randomized rounding algorithm appears higher than that of the deterministic rounding. This seems counterintuitive given the theoretical O(1) approximation for randomized rounding versus the O(k) bound for deterministic rounding. How interpret these empirical results in light of the theoretical guarantees?  For small k, is it possible that the constant hidden in the O(1) bound exceeds k in practice?

4.	What is the scalability of your proposed algorithm for large-scale inputs (e.g., on the order of 100,000 nodes)?

**Ethical Concerns:**

["NO or VERY MINOR ethics concerns only"]

**Final Justification:**

Most of the concerns have been addressed during the rebuttal phase. Overall, considering the topic and contributions, this paper is qualified for an acceptance (a weak accept).

**Limitations:**

There is no potential negative societal impact of their work.

**Paper Formatting Concerns:**

There is no major formatting issues in this paper.

**Quality:**

2

**Strengths And Weaknesses:**

Strengths: The paper introduces a novel framework (LEARN-MEDIAN ) that bridges online learning and learning-augmented algorithms, addressing dynamic metric spaces where instances evolve over time.. The hyperbolic entropy regularizer is an interesting technical enabling dynamic adaptation to expanding metric spaces (e.g., new points appearing online) while maintaining regret guarantees.

Weakness:

1.	The paper only analyzes the average ratio.

2.	The Consistency and Robustness are the keys of a learning-augmented algorithm. This paper did not discuss consistency and robustness.

3.	The paper does not provide the time complexity of the proposed learning-augmented algorithm.

4.	The simulation experiment results are unconvincing. For example, the experiments use synthetic data only; real-world validation would strengthen the claims.

---

> ### Author Rebuttal · Authors · 2025-07-30
>
> We thank the reviewer for the detailed review and we address the weaknesses and questions below:
>
> **R:** The paper focuses on optimizing the average approximation ratio across instances. However, it does not explicitly analyze the algorithm’s Consistency (performance under accurate predictions) or Robustness (worst-case guarantees under prediction errors).
>
> **A:** The definitions of consistency and robustness correspond to algorithms that are given explicit predictions. Our setting is not directly analogous to a prediction-based algorithm (see the first paragraph of related work in the paper), rather we are interested in using the source of predictions, i.e., prior instances of the problem, directly in an online learning framework. Having said that, the notions of consistency and robustness, which capture situations where predictions are good and arbitrarily bad, intuitively correspond to the benchmark (cost of the best solution in hindsight) being small or large in our setting. Note that if the instances are related, then the benchmark cost of a fixed solution is smaller, while if the instances are unrelated, then the benchmark cost of any fixed solution is larger. Our algorithm’s $(\alpha, \beta)$ guarantees hold irrespective of what the value of the benchmark cost is, i.e., the algorithm automatically adapts to whether the instances provide learnable information or not.
>
>
> **R:** The paper does not provide the time complexity of the proposed learning-augmented algorithm. What is the scalability of your proposed algorithm for large-scale inputs (e.g., on the order of $100,000$ nodes)?
>
> **A:** Note that our reduction of each instance to $k$ points helps reduce the running time of the learning algorithm, and makes it independent of the size of the instances $n$. (The $k$-median algorithm has to run on instances of size $n$, but this is unavoidable in any case.) We typically expect that $n$ will be much larger than all other parameters in the problem. Using the reduction, we only need to run our learning algorithm on at most $kT$ points, which is independent of $n$. The time complexity of the learning algorithm is $O(k^2 T^3)$, which comes from needing to iterate through the entries of a distance matrix of size at most $kT \times kT$ every round.
>
> To empirically confirm scalability, we ran experiments with $n= 1000, \ldots, 6000$ and observed that the running time of the learning algorithm was always $2-3$ seconds on Google Colab ($2.20$ GHz Intel® Xeon® CPU, $51$ GB RAM).
>
> **R:** The paper's experiments evaluate the algorithm on a unit square metric space ($[0, 1] \times [0, 1]$). However, in real-world applications, the metric space may have irregular scales
>
> **A:** Note that our theoretical $(\alpha, \beta)$ guarantee holds regardless of the scale of the metric space. As suggested, we ran experiments on $400$ points drawn uniformly from an $(1 \times 10)$ rectangle for $k=2,3,6$ and a time horizon of $T=1000$. In all cases we achieved performance close to the optimal fixed solution in hindsight, with a final approximation ratio of less than $1.3$.
>
> **R:** In Fig. $3$ (for $k = 2, 3, 6$) and 4 (for $k = 4$), the average approximation ratio of the randomized rounding algorithm appears higher than that of the deterministic rounding
>
> **A:** While the randomized algorithm does perform slightly worse than the deterministic algorithm in some cases, it is important to note that the approximation ratios are already quite close to $1$ ($<1.4$) and the difference in performance is only about $10\%$. These results are better interpreted as both algorithms performing very close to the optimal solution, and much better than the worst-case bounds given in the analysis of either algorithm. It is difficult to conjecture on the reasons for the slight differences in performance between the algorithms given they are both so close to the optimum.

---

> > ### Comment · Reviewer_Rx2G · 2025-08-06
> > **Response to the authors**
> >
> > Thank you for your detailed response, which addresses part of my concerns. However, I still have some reservations. Regarding time complexity, current evidence only shows that it can complete in seconds on datasets with the scale of several thousand, while Its scalability to larger datasets remains uncertain especially given the potential overhead of large matrix computations. This paper should also explain in more detail how and why prior instances of the problem can be applied to the learning-augmented framework. At this point, I prefer to keep my initial score.

---

> > > ### Author Response · Authors · 2025-08-07
> > >
> > > We thank the reviewer for the response to our rebuttal and for the additional questions. Regarding scalability, we note that the running time of our online learning algorithm is $O(k^2 T^3)$, which is  independent of the sizes of the instances $n$. Indeed, the dimension of the underlying distance matrix used by our learning algorithm is independent of the instance sizes. So, there are no large matrix computations in our algorithm, given that the size of the matrix is not dependent on the size of the instances. In creating the distance matrix, we use our reduction which employs an approximation algorithm to $k$-median. This approximation algorithm has a running time that scales with the sizes of the instances. However, this is *outside* our online learning algorithm, and *any* off-the-shelf approximation algorithm for $k$-median can be used for this purpose. Furthermore, since the problem being solved is $k$-median, it is unavoidable that some approximation algorithm to $k$-median has to be used.
> > >
> > > Once again, we emphasize that the running time of our online learning algorithm is *independent* of the sizes of the $k$-median instances. We would be happy to answer any further questions in this regard.

---

### Official Review · Reviewer_JsQM · 2025-07-01

**Clarity:** 3
**Significance:** 3
**Originality:** 3
**Rating:** 5
**Confidence:** 3

**Summary:**

The paper explores a learning-augmented model of $k$-median clustering where the learner receives the past problem instances up to the current time point in an online manner and the goal is to use the information from past instances to predict a solution for future instances. The paper proposes a deterministic algorithm that achieves a competitive ratio of $O(k)$ and a regret of $O(\sqrt{T} \log T \log (Tk))$ and a randomized algorithm that achieves a competitive ratio of $O(1)$ and a regret of $O(\sqrt{T} \log T \log (Tk))$. The main framework goes by converting the $k$-median problem to a bounded variant and then calculating a fractional solution and then rounding it to get the final solution. The paper also provides empirical evidence in to the performance of the algorithm. They further strengthen their algorithmic results by showing that in the deterministic setting having competitive ratio of $o(k)$ and regret of $o(T)$ at the same time is impossible (and for randomized setting $(1+\epsilon)$ and $o(T)$ at the same time is impossible).

**Questions:**

1. How well does the algorithm scale with data in practice? Does the bounded variants help with handling the scaling?  How well would it do against a large dataset with small drifts? Or against a large dataset with infrequent but large drifts?

2. What are the key technical barriers in extending your framework to non-metric problems?

3. How useful might the paradigms such as side-information be in this learning-augmented setting? For instance if there is some information about some optimal centers or some approximate information, would that be useful? What information might be useful in breaking the current lower bounds?

4. How sensitive is your method to noisy or outlier instances? How well would the algorithm do if some $\alpha$ fraction of the instances are noisy or outliers? Would it still give the same guarantees? Is there any specific steps to take to make it robust against such cases? Or is it inherently robust?

**Ethical Concerns:**

["NO or VERY MINOR ethics concerns only"]

**Final Justification:**

The authors clarified the concerns I had on the model, scalability and adaptability. Therefore, I will retain my rating.

**Limitations:**

Yes

**Quality:**

3

**Strengths And Weaknesses:**

Strengths

1. The paper explores the $k$-median clustering problem using the notions of learning-augmented algorithms and online learning. Unlike in standard online $k$-clustering with data arriving one by one, we have new instances arriving and past instances giving hints on how the new instances might be solved. This formulation would be useful specially in handling things such as evolving data.

2. The paper provides algorithms with provable cost guarantees and strengthens their work by providing nearly similar lower bounds.

3. The algorithm and the proofs are well structured and easy to follow. The paper is overall well written.


Weaknesses

1. The motivation for the specific formulation of the problem is not entirely clear at the start. I can see why the past instances informing on future instances would be an interesting formulation in settings where there is evolving data. It might be nice to clarify why this model is interesting (maybe using some potential applications).

2. The experiments seems to be limited to synthetic data. It would strengthen the empirical claims to include real-world datasets where dynamic clustering is relevant.

---

> ### Author Rebuttal · Authors · 2025-07-30
>
> We thank the reviewer for the detailed review and we address the weaknesses and questions below:
>
> **R:** It might be nice to clarify why this model is interesting (maybe using some potential applications).
>
> **A:** We give several applications that we will add to the paper:
>
> $\bullet$ Developing accurate methods to cluster evolving data streams has been identified as an open challenge in data stream mining research, e.g.:
>
> *Georg  Krempl, Indre Zliobaite, Dariusz Brzezinski, Eyke Hüllermeier, Mark Last, Vincent Lemaire, Tino Noack, Ammar Shaker, Sonja Sievi, Myra Spiliopoulou, Jerzy Stefanowski. Open challenges for data stream mining research. ACM SIGKDD Explorations Newsletter, 2014.*
>
> $\bullet$ Aggregating similar news stories based on their similarities can be used for recommendations downstream, e.g.:
>
> *Shufeng Gong, Yanfeng Zhang, Ge Yu. Clustering Stream Data by Exploring the Evolution of Density Mountain. Proceedings of the VLDB Endowment, 2017.*
>
> This paper studies news articles that are inherently dynamic and evolve over time as fresh stories emerge, and older, less relevant news fades out. To provide timely and accurate recommendations, a clustering algorithm that adapts to evolving data is utilized.
>
> $\bullet$ Clustering sensor data, e.g.:
>
>
> *Zhengru Wang, Xin Wang, Shuhao Zhan. MOStream: A Modular and Self-Optimizing Data Stream Clustering Algorithm.  IEEE International Conference on Data Mining (ICDM) 2024.*
>
> This paper studies sensor data in the form of evolving data streams. The data depends on environmental conditions that can change dramatically between day and night or due to localized events such as forest fires, as well as fluctuate significantly across seasons. The goal is to maintain a clustering of such data as it evolves over time.
>
> $\bullet$ Clustering network traffic, e.g.:
>
> *Feng Cao, Martin Ester, Weining Qian, Aoying Zhou. Density-Based Clustering over an Evolving Data Stream with Noise. SIAM International Conference on Data Mining (SDM) 2006.*
>
> The last three references also study clustering of network traffic data, where patterns of network user connections change slowly over time in normal circumstances, but can also undergo sudden shifts if a malicious attack occurs. Consequently, network-monitoring methods that use clustering need to be able to adapt to shifts in network traffic and these shifts can be in the form of gradual drifts or adversarial changes.
>
>
>
> **R:** How well does the algorithm scale with data in practice? Does the bounded variants help with handling the scaling?
>
> **A:** Note that our reduction of each instance to $k$ points helps reduce the running time of the learning algorithm, and makes it independent of the size of the instances $n$. (The $k$-median algorithm has to run on instances of size $n$, but this is unavoidable in any case.) We typically expect that $n$ will be much larger than all other parameters in the problem. Using the reduction, we only need to run our learning algorithm on at most $kT$ points, which is independent of $n$. The time complexity of the learning algorithm is $O(k^2 T^3)$, which comes from needing to iterate through the entries of a distance matrix of size at most $kT \times kT$ every round.
>
> To empirically confirm scalability, we ran experiments with $n= 1000, \ldots, 6000$ and observed that the running time of the learning algorithm was always $2-3$ seconds on Google Colab ($2.20$ GHz  Intel® Xeon® CPU, $51$ GB RAM).
>
>
> **R:** How well would it do against a large dataset with small drifts? Or against a large dataset with infrequent but large drifts?
>
> **A:** The experiment named “Scale Changes” in our paper analyzes the performance of our algorithm when the incoming instances have sudden shifts. In this experiment, we have 5 clusters with 10 points each and the clusters grow geometrically far apart in distance. The sequence of instances iterates over these clusters in batches of increasing length. We observed that our algorithm quickly adapts to the changing environment and shifts a center to the new cluster.
>
> As suggested by the reviewer, we also ran some experiments to analyze the behavior of our algorithm when instances drift slowly over time. In these experiments, we have $n=2500, T= 250$, and we use a range of values for $k = 1,2,3$. The underlying instances are created by randomly drawing $5$ points in a disc around the origin, but then shifting the center to the right by $0.02$ for each instance. The final average approximation ratios were less than $2$ in all cases. (In contrast, if the algorithm were to use the initial solutions throughout, i.e., does not automatically adapt to the drift, then the cost ratios are around $5-15$.)
>
>
>
> **R:** What are the key technical barriers in extending your framework to non-metric problems?
>
> **A:**  Incorporating online learning based techniques to improve the solution for online non-metric clustering problems is an interesting question. However, we note that in the non-metric setting, it is not possible to compete with the optimal solution in hindsight. This is because the last instance can reveal the optimal solution in hindsight, which is close to all previous instances that are mutually far from each other. Since the online learning algorithm only gets to see these optimal centers at the very end, it cannot choose a competitive solution for the prior instances. Therefore, in the non-metric setting, one would need to design a new benchmark against which an online learning algorithm can compete across a sequence of instances.
>
> **R:** How useful might the paradigms such as side-information be in this learning-augmented setting?
>
> **A:** Introducing side information is an interesting direction to consider. For instance, suitable side information may help in reducing the regret (the $\beta$ in our guarantee) by providing a good starting solution for the mirror descent algorithm. However, this would require a completely new analysis which is beyond the scope of the current work.
>
> **R:** How sensitive is your method to noisy or outlier instances?
>
> **A:** The effect of outliers is another interesting direction. First, we note that our algorithm is inherently robust in the sense that the $(\alpha,\beta)$ approximation guarantee holds even in the presence of outliers. That said, the outliers degrade the benchmark since there is no good solution in hindsight that serves all instances including the outliers. If we wanted to compete against the best solution that is allowed to disregard the outlier instances, then we would likely need to redesign our algorithm completely.

---

> > ### Comment · Reviewer_JsQM · 2025-08-04
> >
> > Thank you for the detailed response to my questions. Based on the answers, I will retain my rating as is.

---

### Official Review · Reviewer_v3YS · 2025-07-02

**Clarity:** 3
**Significance:** 3
**Originality:** 3
**Rating:** 4
**Confidence:** 3

**Summary:**

This paper studies an online learning setting of the k-median problem. The setting considers a sequence of instances, each corresponding to a classic k-median problem. Before the realization of each instance, the algorithm must select k centers. Its performance is measured by the cost ratio between the algorithm's solution (i.e., the total distance from points to their nearest centers) and the offline optimal solution that has access to all data in advance. The goal is to achieve an $(\alpha,\beta)$-approximation with respect to the cumulative cost ratio, where $\alpha$ denotes the multiplicative factor and $\beta$ is the additive term.
The paper proposes two algorithms:
- A deterministic algorithm with $\alpha = O(k)$ and $\beta = o(T)$
- A randomized algorithm with $\alpha = O(1)$ and $\beta = o(T)$

The paper also prove that these results match the lower bounds.

The proposed approach begins by deriving an online fractional solution to a reduced version of the original problem that involves only k points. The final algorithms (deterministic or randomized) are online rounding schemes applied to this fractional solution. The paper also includes experiments on synthetic data to illustrate performance.

**Questions:**

- Could you provide a concrete application scenario for the online learning setting over multiple k-median instances? In particular, (i) Why is an online setting necessary? (ii) Why is the cost ratio of each instance more relevant than the absolute cost?

- How does this work compare to the data-driven algorithm design framework in the online setting? What are the key differences, and why wouldn't existing techniques from that literature apply here?

- A key step in the algorithm is to reduce the original problem to one with only k points. Why is exactly k chosen? Would it be possible (or even beneficial) to reduce the problem to fewer or slightly more than k points? How would this affect performance or complexity?

**Ethical Concerns:**

["NO or VERY MINOR ethics concerns only"]

**Final Justification:**

The authors partially address my concerns about (i) motivating scenarios of the online setting, (ii) choice of performance metrics, and (iii) its connection to data-driven algorithm design. But those will not change my rating of the paper.

**Quality:**

4

**Strengths And Weaknesses:**

Strength
- The proposed algorithms achieve tight bounds on both multiplicative and additive approximation factors for deterministic and randomized algorithms.

- The paper presents several nice ideas, including: (i) reducing the original problem to a smaller instance with only k points to manage computational complexity, and
(ii) using a novel regularizer to address the challenge of an expanding solution space in the online setting.


Weakness
- The motivation behind the performance metric could be clearer. The $(\alpha,\beta)$-approximation is defined on the cumulative cost ratios, which is less standard compared to bounding the cumulative cost directly. Additionally, the benchmark used is static, which is a bit weak.

- Although the paper refers to learning-augmented algorithms, it more closely aligns with the data-driven algorithm design literature (e.g., M. Balcan, Data-Driven Algorithm Design, 2020). This is essentially an online learning problem over multiple rounds, where each round presents a new k-median instance. The paper lacks a discussion connecting to this line of research.

---

> ### Author Rebuttal · Authors · 2025-07-30
>
> We thank the reviewer for the detailed review and we address the weaknesses and questions below.
>
> **R:** Could you provide a concrete application scenario for the online learning setting over multiple k-median instances?
>
> **A:** We give several applications that we will add to the paper:
>
> $\bullet$ Developing accurate methods to cluster evolving data streams has been identified as an open challenge in data stream mining research, e.g.:
>
> *Georg  Krempl, Indre Zliobaite, Dariusz Brzezinski, Eyke Hüllermeier, Mark Last, Vincent Lemaire, Tino Noack, Ammar Shaker, Sonja Sievi, Myra Spiliopoulou, Jerzy Stefanowski. Open challenges for data stream mining research. ACM SIGKDD Explorations Newsletter, 2014.*
>
> $\bullet$ Aggregating similar news stories based on their similarities can be used for recommendations downstream, e.g.:
>
> *Shufeng Gong, Yanfeng Zhang, Ge Yu. Clustering Stream Data by Exploring the Evolution of Density Mountain. Proceedings of the VLDB Endowment, 2017.*
>
> This paper studies news articles that are inherently dynamic and evolve over time as fresh stories emerge, and older, less relevant news fades out. To provide timely and accurate recommendations, a clustering algorithm that adapts to evolving data is utilized.
>
> $\bullet$ Clustering sensor data, e.g.:
>
>
> *Zhengru Wang, Xin Wang, Shuhao Zhan. MOStream: A Modular and Self-Optimizing Data Stream Clustering Algorithm.  IEEE International Conference on Data Mining (ICDM) 2024.*
>
> This paper studies sensor data in the form of evolving data streams. The data depends on environmental conditions that can change dramatically between day and night or due to localized events such as forest fires, as well as fluctuate significantly across seasons. The goal is to maintain a clustering of such data as it evolves over time.
>
> $\bullet$ Clustering network traffic, e.g.:
>
> *Feng Cao, Martin Ester, Weining Qian, Aoying Zhou. Density-Based Clustering over an Evolving Data Stream with Noise. SIAM International Conference on Data Mining (SDM) 2006.*
>
> The last three references also study clustering of network traffic data, where patterns of network user connections change slowly over time in normal circumstances, but can also undergo sudden shifts if a malicious attack occurs. Consequently, network-monitoring methods that use clustering need to be able to adapt to shifts in network traffic and these shifts can be in the form of gradual drifts or adversarial changes.
>
>
>
> **R:** Why is the cost ratio of each instance more relevant than the absolute cost?
>
> **A:** We considered cost ratios instead of the absolute cost as our performance metric for two reasons.
>
> First, the instances that arrive can be at very different scales. In particular, if one instance is very large and the rest are small, it suffices to perform well on just the large instance when looking at absolute cost. Thus, taking cost ratios ensures that the algorithm is incentivized to do well on every instance, regardless of size.
>
> Second, considering cost ratios instead of absolute costs is not only the right choice from a modeling perspective, but is also necessary for obtaining any regret bounds independent of the number of points $n$. To see this, suppose we get a very large instance with essentially all $n$ points that is far away from all the previous (much smaller) instances which are cumulatively on $o(n)$ points. Any online learning algorithm would then incur a cost of $n \Delta$ for this single instance. On the other hand, the benchmark solution can simply optimize for this large instance and incur small cost for this instance. It suffers a large cost on the previous instances, but these instances being cumulatively small (only $o(n)$ points), the overall cost of the benchmark remains small in terms of absolute cost.
>
> **R:** Additionally, the benchmark used is static, which is a bit weak.
>
> **A:** The benchmark considered is the best fixed solution in hindsight, which is standard in online learning. As in online learning, no algorithm can be competitive with a benchmark that uses a separate solution for each instance. Such a benchmark would have a cost ratio of $1$ for every instance, i.e., would be optimal for every instance. In contrast, an algorithm trained on prior instances can have an arbitrary approximation bound on the next instance. For example, suppose $k=1$ and we receive a concentrated cluster of points in each instance that is far away from all previous instances. An algorithm trained on prior instances cannot predict where the new instance is going to be and hence will have arbitrarily large cost on it, whereas an optimal solution specifically for this instance will have a cost ratio of $1$. This is not specific to our model, but a general feature of any online learning setting, see e.g., Chapter 10 in [Hazan 2016].
>
> **R:** How does this work compare to the data-driven algorithm design framework in the online setting?
>
> **A:** In the data-driven paradigm, the typical goal is to pick the best algorithm out of a palette of algorithms whose behavior is dictated by a numerical parameter, see e.g., the survey:
>
> *Maria-Florina Balcan. Data-driven Algorithm Design. Chapter 29, Beyond the Worst-Case Analysis of Algorithms. Cambridge University Press.*
>
> In this line of work, the typical assumption is that the instances are drawn from an unknown distribution, and the goal is to obtain sample complexity bounds for a given target average performance in comparison to the best parameter choice. The analysis usually proceeds by obtaining bounds on the pseudo-dimension of the cost functions corresponding to the parametrized algorithms, which are then converted to sample complexity bounds using statistical learning tools.
>
> This is in contrast to our setting where the algorithm is allowed to output any valid solution in each round (i.e., there is no parameter choice being made, rather an entire solution is being constructed) and there is no distributional assumption on the sequence of instances. Our techniques leverage ideas from online learning (as against statistical learning) and the final results are about average performance across the adversarial set of instances, rather than expected performance on a distribution.
>
> There are two papers in the data-driven literature that we are aware of which have an online learning flavor. One paper considers an online learning model similar to ours (reference [*Khodak et al. 2022*] in our paper):
>
> [*Khodak et al. 2022*] *Misha Khodak, Maria-Florina Balcan, Ameet Talwalkar, and Sergei Vassilvitskii. Learning predictions for algorithms with predictions. NeurIPS 2022.*
>
> The main difference between this work and ours is that they use prior instances to learn a prediction for the current round, while we use the instances directly to learn a solution for the current round. Moreover, they consider problems such as ski rental, bipartite matching, and scheduling, while we focus on the $k$-median clustering problem.
>
> A second paper that considers the parameterized framework but in an online learning setting (i.e., not distributional) is:
>
> *Maria-Florina Balcan, Travis Dick, and Ellen Vitercik. Dispersion for Data-Driven Algorithm Design, Online Learning, and Private Optimization. FOCS 2018.*
>
> In this paper, instances arrive over time and we pick a parameter for each (that corresponds to an algorithm) and see the corresponding cost. The benchmark in this case is the algorithm corresponding to the best fixed parameter in hindsight. The techniques heavily utilize the parametrization of the algorithms, and use properties that dictate the discontinuity of the cost functions such as piecewise Lipschitz-ness and dispersion (a measure of how many discontinuities are present in a bounded region) that are not directly applicable in our setting.
>
> We will include a detailed comparison with the data-driven literature in our paper.
>
> **R:** A key step in the algorithm is to reduce the original problem to one with only $k$ points. Why is exactly $k$ chosen?
>
> **A:** This reduction eliminates the dependence on the size of instances $n$ in our regret bounds. Otherwise, the subgradient that we compute at each step can then have a large $L_\infty$ norm that depends on $n$, which would then affect the regret term. On the other hand, if we apply our mirror descent algorithm to only $k$ points in each round, then the total number of points under consideration is at most $kT$. Thus, the regret term is independent of $n$.
>
> A second benefit of the reduction is that it reduces the running time of the learning algorithm, and makes it independent of the size of the instances $n$. (The $k$-median algorithm has to run on instances of size $n$, but this is unavoidable in any case.) We typically expect that $n$ will be much larger than all other parameters in the problem. Using the reduction, we only need to run our learning algorithm on at most $kT$ points, which is independent of $n$. (The time complexity of the learning algorithm is $O(k^2 T^3)$, which comes from needing to iterate through the entries of a distance matrix of size at most $kT \times kT$ every round.) To empirically confirm scalability, we ran experiments with $n= 1000, \ldots, 6000$ and observed that the running time of the learning algorithm was always $2-3$ seconds on Google Colab ($2.20$ GHz  Intel® Xeon® CPU, $51$ GB RAM).
>
> Finally, we comment that it is important to choose exactly $k$ points in our reduction. If we use fewer or more than $k$ points, then the weights for the reduced instance are inconsistent with the $k$-median optimum. E.g., if using $< k$ points and there are $k$ closely knit clusters, then the resulting cost of the solution (which determines weights) is arbitrarily large. Similarly, if we use $> k$ points and there are $k+1$ closely knit clusters, then the resulting cost of the solution is arbitrarily small.
>
> We will include this discussion in the paper.

---

> > ### Comment · Reviewer_v3YS · 2025-08-03
> >
> > Thanks for your detailed response addressing my concerns. I will keep my rating.

---

### Official Review · Reviewer_vt4Z · 2025-07-02

**Clarity:** 2
**Significance:** 2
**Originality:** 1
**Rating:** 4
**Confidence:** 4

**Summary:**

This paper addresses the k-median clustering problem in an online setting. The input is an online sequence of instances (point sets in a metric space) $V_0, V_1, V_2, \ldots, V_T$ that arrive one by one. At any time point $t \leq T$, it is assumed that the algorithm has access to the prior instances $V_0, V_1, V_2, \ldots, V_{t-1}$. The algorithm uses the knowledge of these prior instances to compute a k-median solution $Y_t$, which is then applied to instance $V_t$ and evaluated for loss. The overall goal is to minimize the total loss over the entire sequence, i.e., to minimize $\sum_{t} P(Y_t, V_t)$, where $P(Y, V)$ denotes the cost of using center set $Y$ on point set $V$. The main result of the paper is an $(\alpha, \beta)$-approximation to the optimal solution, which is defined with respect to the so-called “best fixed solution in hindsight.” Let $Y^*$ be the set of k centers that minimizes $\sum_{t} P(Y^*, V_t)$. The proposed algorithm computes $Y_t$ for all $t$, such that $\sum_t P(Y_t, V_t) \leq \alpha \sum_{t} P(Y^*, V_t) + \beta$,
where $\alpha = O(1)$ and $\beta = o(T)$, and $\beta$ depends polynomially on the spread ratio $\Delta$ and $k$.

**Questions:**

Questions:

Is there justification that the method indeed learn from previous instances?

What is the justification of several problem settings. For example, what is the justification for evaluating the cost of $Y_t$ on $V_t$ only, rather than over a cumulative set (e.g., $V_0, \ldots, V_t$)?

**Ethical Concerns:**

["NO or VERY MINOR ethics concerns only"]

**Final Justification:**

The authors have addressed all my concerns in their rebuttal and during the communication period. For this reason, I have raised the score from 2 to 4.

**Limitations:**

Yes

**Paper Formatting Concerns:**

No major formatting issue has been identified.

**Quality:**

2

**Strengths And Weaknesses:**

Strengths:

The paper addresses a new variant of the k-median problem in an online setting with theoretical guarantee.


Weaknesses:

It could be misleading to claim the result as learning-augmented. From the perspective of learning-augmented algorithms, I do not believe the result really captures the spirit of "applying machine learning techniques on prior problem instances to inform algorithmic decisions for a future instance." In fact, the proposed algorithm makes no assumptions about the structure or distribution of the prior instances $V_i$. In an extreme case, even if the $V_i$'s are entirely unrelated—so that, in principle, no useful information can be learned from earlier instances to aid future ones—the $(\alpha, \beta)$-approximation guarantee still holds (!). This suggests that the theoretical guarantee presented in the paper does not stem from any learning from prior instances.

As a result, the contribution seems more accurately classified as an analysis of a certain class of online algorithms, rather than as a learning-augmented result. Specifically, it addresses the question: if at each time step $t$, the algorithm is restricted to accessing only the previous instances $V_0, V_1, \ldots, V_{t-1}$ to compute $Y_t$, how can we minimize the total cost of using $Y_t$ on $V_t$?

However, even if we evaluate the work purely as a theoretical result—without the learning-augmented perspective—the problem formulation still seems confusing. It is unclear why the objective focuses on the cost of $Y_t$ with respect to $V_t$, rather than, say, with respect to $V_0, \ldots, V_t$. Moreover, the definition of the optimal solution $Y^*$ is somewhat puzzling: it does not make sense to me to use a fixed $Y^*$ for all $t$ as the optimal solution. It seems more natural to define an optimal solution $Y^*_t$ for each time $t$, and then compare $Y_t$ to $Y^*_t$.

Presentation can be improved: The algorithm description in Chapter 3 lacks a high-level description of the main idea, making it difficult to follow

Experiment: Relies solely on synthetically data (however, given the aforementioned issues with the algorithm's ability to learn from data, I would suspect the use of real-world data would likely make little difference in the results). The experimental section also lacks a clear summary about the implications of the experiment results.

---

> ### Author Rebuttal · Authors · 2025-07-30
>
> We thank the reviewer for the detailed review. We address the questions below.
>
> **R:** It could be misleading to claim the result as learning-augmented.... This suggests that the theoretical guarantee presented in the paper does not stem from any learning from prior instances.
>
> **A:** Our framework is based on the well-studied online learning framework; see the standard textbook below (reference [*Hazan 2016*] in our paper):
>
> [*Hazan 2016*] *Elad Hazan. Introduction to online convex optimization. Foundations and Trends in Optimization.*
>
> Online learning has a vast literature with no distributional assumptions on the inputs across different rounds (chapters 1-5, 7, 10 in [*Hazan 2016*] and references contained therein). As in online learning, our paper makes no distributional assumption either.  In online learning, the algorithm chooses an action for the (unknown) instance in the current round based on the outcome for prior rounds. Similarly, in our paper, the algorithm chooses a $k$-median clustering solution for the (unknown) instance in the current round based on the instances in the prior rounds. Indeed, learning-augmented algorithms in the online learning framework have been considered in prior work (reference [*Khodak et al. 2022*] in our paper):
>
> [*Khodak et al. 2022*] *Misha Khodak, Maria-Florina Balcan, Ameet Talwalkar, and Sergei Vassilvitskii. Learning predictions for algorithms with predictions. NeurIPS 2022.*
>
> The main difference between this work and ours is that they use prior instances to learn a prediction for the current round, while we use the instances directly to learn a solution for the current round. Moreover, they consider problems such as ski rental, bipartite matching, and scheduling, while we focus on the $k$-median clustering problem.
>
> In [*Khodak et al. 2022*], as well as in online learning in general, a sequence of instances are presented to the algorithm (with no distributional assumption), and it must produce a solution for the next instance based on the prior instances. The goal is to compete with the best fixed solution in hindsight. This is exactly the setting that we consider in our paper. The intuition in online learning is that if the instances are related, then the benchmark cost (of the best fixed solution) is lower, while if the instances are unrelated, then the benchmark cost is higher. The $(\alpha, \beta)$ guarantee holds for any sequence of instances, but in the former case, it leads to a smaller cost bound for the algorithm while in the latter case, it leads to a larger cost bound. Again, this is the same as in the entire literature in online learning.
>
> We emphasize that our setting is not directly analogous to a prediction-based learning-augmented algorithm (we mention this in the first paragraph of related work in the paper). Indeed, in many papers using explicit predictions, the source of the prediction is vaguely attributed to prior instances, e.g., the survey (reference [Mitzenmacher and Vassilvitskii 2020] in our paper):
>
> [*Mitzenmacher and Vassilvitskii 2020*] *Michael Mitzenmacher and Sergei Vassilvitskii. Algorithms with predictions. Chapter 30, Beyond the Worst-Case Analysis of Algorithms. Cambridge University Press.*
>
> and other works such as these references from our paper, including for clustering problems such as $k$-means and facility location:
>
>
> [*Ergun et al. 2022*] *Jon C. Ergun, Zhili Feng, Sandeep Silwal, David P. Woodruff, and Samson Zhou. Learning-augmented k-means clustering. ICLR 2022.*
>
> [*Jiang et al. 2022*] *Shaofeng H-C. Jiang, Erzhi Liu, You Lyu, Zhihao Gavin Tang, and Yubo Zhang. Online facility location with predictions. ICLR 2022.*
>
> [*Lykouris and Vassilvitskii 2021*] *Thodoris Lykouris and Sergei Vassilvitskii. Competitive caching with machine learned advice. Journal of the ACM, 2021.*
>
> In our paper, we make the source and role of side information (such as predictions) more transparent by showing that we can use online learning on prior instances of the $k$-median problem to inform solutions for future instances of the problem.
>
> **R:** Moreover, the definition of the optimal solution $Y^\star$ is somewhat puzzling: it does not make sense to use a fixed $Y^*$ for all $t$ as the optimal solution. It seems more natural to define an optimal solution $Y_t^\star$ for each time $t$, and then compare $Y_t$ to $Y_t^\star$.
>
> **A:** The benchmark considered is the best fixed solution in hindsight, which is standard in online learning. As in online learning, no algorithm can be competitive with a benchmark that uses a separate solution for each instance. Such a benchmark would have a cost ratio of $1$ for every instance, i.e., would be optimal for every instance. In contrast, an algorithm trained on prior instances can have an arbitrary approximation bound on the next instance. For example, suppose $k=1$ and we receive a concentrated cluster of points in each instance that is far away from all previous instances. An algorithm trained on prior instances cannot predict where the new instance is going to be and hence will have arbitrarily large cost on it, whereas an optimal solution specifically for this instance will have a cost ratio of $1$. This is not specific to our model, but a general feature of any online learning setting, see e.g., Chapter 10 in [*Hazan 2016*].
>
>
>
> **R:** What is the justification for evaluating the cost of  $Y_t$  on $V_t$ only, rather than over a cumulative set (e.g., $V_0, …, V_t$)?
>
>
> **A:** We evaluate the cost of the solution on $V_t$ instead of the historical instances because the algorithm gives the solution $Y_t$ specifically for the next instance $V_t$. This solution is not intended to be used for prior instances $V_0, V_1, \ldots, V_{t-1}$ that are already in the past at time $t$. This setup is standard in the online learning literature.
>
>
> **R:** Presentation can be improved: The algorithm description in Chapter 3 lacks a high-level description of the main idea, making it difficult to follow. The experimental section also lacks a clear summary about the implications of the experiment results.
>
>
> **A:** We will add details and intuition giving a high-level overview of the algorithm. The main takeaways from the experiments are described at the end of Section 1.1. We will also add a summary of the takeaways from the numerical simulations in the experimental section.

---

> > ### Comment · Reviewer_vt4Z · 2025-08-03
> > **Responses to authors' rebuttal**
> >
> > I thank the authors for the clarification provided on how the proposed algorithm fits into the online learning literature, especially the connection to [Khodak et al. 2022] (while the authors of this work did not claim it is a learning-augmented in the paper, it is recognized in some later works as learning-augmented). The connection to previous work is now clearer, and I will raise my score to "borderline accept".
> >
> > That said, I still have some concerns. The proposed $(\alpha,\beta)$ bound seems to merely adapt to how “easy” a sequence is (i.e., smaller benchmark cost when instances are related). The information from past instances influence future predictions in a very implicit way. The paper does not demonstrate the explicit mechanism by which the solution learned from past instances is truly informing the predication of later instances, beyond what a similar online learning algorithm would inherently achieve.
> >
> > Since this paper has emphasized the algorithm’s capabilities to use past instances to inform future decisions, it would be better to provide more explicit evidence and justification in the paper.

---

> > > ### Author Response · Authors · 2025-08-04
> > >
> > > We thank reviewer vt4Z for the positive assessment of our work post rebuttal. We also appreciate the question about the mechanism by which prior instances influence later solutions. We note that a black box online learning algorithm is not usable in our context, since it needs to know the space of solutions up front. Therefore, while the framework is using concepts from online learning, the algorithm that we propose is novel. Indeed, we see this algorithm itself as the explicit mechanism by which the prior instances are influencing the solution for future instances. Our experiments also make this apparent - for instance, in all the experiments reported in Fig. 2, the key observation is that the algorithm changes the solution for a future instance in response to a changing sequence of prior instances. We will clarify this connection between prior instances and future solutions both theoretically and empirically further in the paper.

---

> > > > ### Comment · Reviewer_vt4Z · 2025-08-06
> > > >
> > > > Thanks for the clarification. I do not have any further questions. As mentioned in previous comments, I will raise the score to "borderline accept".

---

### Note · Authors · 2025-08-11

We thank all the reviewers for the detailed reviews and follow up discussions. We will add the new points that came up during the discussions to the paper and revise the writing for improved clarity.

---

### Decision · Program_Chairs · 2025-09-17

**Decision:**

Accept (poster)

**Comment:**

This paper addresses the k-median clustering problem in an online setting. The input is an online sequence of T instances that arrive one by one. At any point in time, it is assumed that the algorithm has access to the prior instances. The algorithm uses the knowledge of these prior instances to compute a k-median solution, which is then applied to the current instance and evaluated for loss. The overall goal is to minimize the total loss over the entire sequence. The main result of the paper is an (alpha,beta)-approximation of the optimal solution, which is defined with respect to the so-called “best fixed solution in hindsight.” This means that the total loss occurred is at most alpha times the loss of the best fixed solution plus beta. A deterministic algorithm with alpha=O(k) and beta=o(T) is shown and a randomized one with alpha=O(1) and beta=o(T). Also lower bounds are presented showing that these results are tight.

All reviewers think that the results are interesting and non-trivial. The submission is also well-written. Most of the weaknesses mentioned by the reviewers concern the motivation of the model and the connection to other models and results in the literature (e.g. data-driven algorithms). All these questions could be addressed convincingly by the authors, and all reviewers are positive about the submission. I also agree that the submission presents interesting and non-trivial results and that the considered problem is well motivated and important.